# Adaptive multi-temperature control for transport and storage containers enabled by phase-change materials

Xinchen Zhou [1,2,3], Xiang Xu [4,5] & Jiping Huang [1,2,3] ✉

The transportation of essential items, such as food and vaccines, often requires adaptive multi-temperature control to maintain high safety and efficiency. While existing methods utilizing phase change materials have shown promise, challenges related to heat transfer and materials' physicochemical properties remain. In this study, we present an adaptive multi-temperature control system using liquid-solid phase transitions to achieve highly effective thermal management using a pair of heat and cold sources. By leveraging the properties of stearic acid and distilled water, we fabricated a multi-temperature maintenance container and demonstrated temperature variations of only 0.14-2.05% over a two-hour period, underscoring the efficacy of our approach. Our findings offer a practical solution to address critical challenges in reliable transportation of goods, with potential implications for various fields in physical, engineering, and life sciences.

Implementing multi-temperature control systems is crucial for maintaining high efficiency in various critical domains such as goods transportation[1], cold chain logistics[2–4], battery thermal management[5], building environment[6], and thermal energy storage[7–11]. Cutting-edge technologies, utilizing multiple phase-change materials (PCMs) as heat/cold sources with advantages in energy storage and mobility, have considerable potential in achieving this aim by controlling one zone per PCM[4,5,8–11]. A number of optimization studies[5–11] have underlined the significance of not only examining the performance of the heat and cold sources in isolation but also scrutinizing system design parameters. This brings to light the necessity for developing general methods to bolster the efficiency of multi-temperature control systems.

To illustrate the importance of multi-temperature control, let us contemplate goods transportation. It is widely recognized that virus epidemics and disasters considerably limit human activities[12–14]. Therefore, to ensure the fulfillment of essential human needs and sustain social stability, efficient and secure

transportation of various goods, such as food and vaccines, is vital whilst considering the dynamic supply-demand relationship[15–23]. Significantly, these goods may possess differing temperature requirements (Fig. 1a), which can provoke heat transfer among them. These temperature interactions necessitate a multi-temperature control approach during transportation to uphold the desired temperature conditions. By employing such an approach, reliable and secure transportation of crucial resources can be accomplished, thereby preserving their integrity and optimizing energy efficiency.

While multi-temperature control has been developed in cold chain logistics[4,24–27], goods transportation presents challenges due to its wider temperature range. Nonetheless, we can glean insights from the existing cold chain logistics practices. In cold chain logistics, there exist two schemes to actualize multi-temperature control:[24–27] multi-vehicle distribution and multi-temperature joint distribution. The former employs various vehicles with differing temperature storage spaces, whilst the

¹Department of Physics, Fudan University, Shanghai 200438, China. ²State Key Laboratory of Surface Physics, Fudan University, Shanghai 200438, China. ³Key Laboratory of Micro and Nano Photonic Structures (MOE), Fudan University, Shanghai 200438, China. ⁴Key Lab of Smart Prevention and Mitigation of Civil Engineering Disasters of the Ministry of Industry and Information Technology, Harbin Institute of Technology, Harbin 150090, China. ⁵Key Lab of Structures Dynamic Behavior and Control of the Ministry of Education, Harbin Institute of Technology, Harbin 150090, China. ✉e-mail: jphuang@fudan.edu.cn

latter designates multiple storage spaces to diverse temperatures within a single vehicle. Prior research has suggested that multi-temperature joint distribution is more efficient and cost-effective than multi-vehicle distribution[24,25]. Thus, given the critical importance of transporting goods during an epidemic, the multi-temperature joint distribution represents a feasible solution. However, the need for efficiency and convenience in transportation restricts the application of some temperature control technologies that necessitate additional mechanical equipment or real-time external energy input, such as vapor compression refrigeration and electric heating.

As aforementioned, phase-change technology holds potential in this scenario due to its advantages in energy storage characteristics, easy operation, simple structure, and low cost[4,18-21,28-30]. By employing PCM to absorb or release substantial latent heat during phase transition, this technology can effectively control the temperature of specific areas at the phase transition temperature (Fig. 1b)[2,4,21,24-27]. Nevertheless, it also presents two obstacles: (1) temperature interaction among the spaces storing goods with differing temperature requirements; (2) physicochemical properties of PCM. For the first obstacle, existing multi-temperature control techniques minimize temperature interactions by enhancing the thermal insulation in between[4,31]. This suggests that when the temperature differences among the storage spaces are significant, more energy may be employed to negate the negative effects of temperature interactions. Moreover, a specific type of PCM can only strictly control the temperature of a single zone[19-21,28-30]. Placing various types of PCMs with

differing phase transition temperatures in multiple zones to actualize multi-temperature control for goods with different temperature requirements occupies much space, thus diminishing space utilization. The second obstacle demands the phase transition temperature of the PCM to be akin to the temperature requirement of the good. It further requires the PCM to possess fairly comprehensive properties, encompassing thermal properties, thermal stability, chemical stability, cycling stability, and economic efficiency[19-21,30,32,33]. For various goods with differing temperature requirements, if there is no PCM meeting the requirements in nature, it becomes necessary to develop multiple composite PCMs, necessitating a substantial initial investment. The inherent flaws of some materials, such as supercooling and phase separation, further escalate the development difficulty[34-38]. Regarding these two obstacles, our group has been devoted to the thermal/structural parameters design in macroscopic heat transfer systems for a long period and has realized various thermal functions[39-51]. One notable approach is the precise regulation of heat flux to maintain a constant temperature amidst varying thermal gradients, as opposed to relying solely on conventional thermal insulation[41]. In view of the rapid development of metamaterials in diverse fields[46-49,52-58], attaining efficient multi-temperature control whilst aptly utilizing temperature interactions through artificial design might herald exciting advancements in goods transportation.

We draw inspiration from agricultural production techniques that utilize terraces with multiple flat areas within a height difference. This similar structure can be applied to accomplish efficient multi-temperature control in goods transportation. As

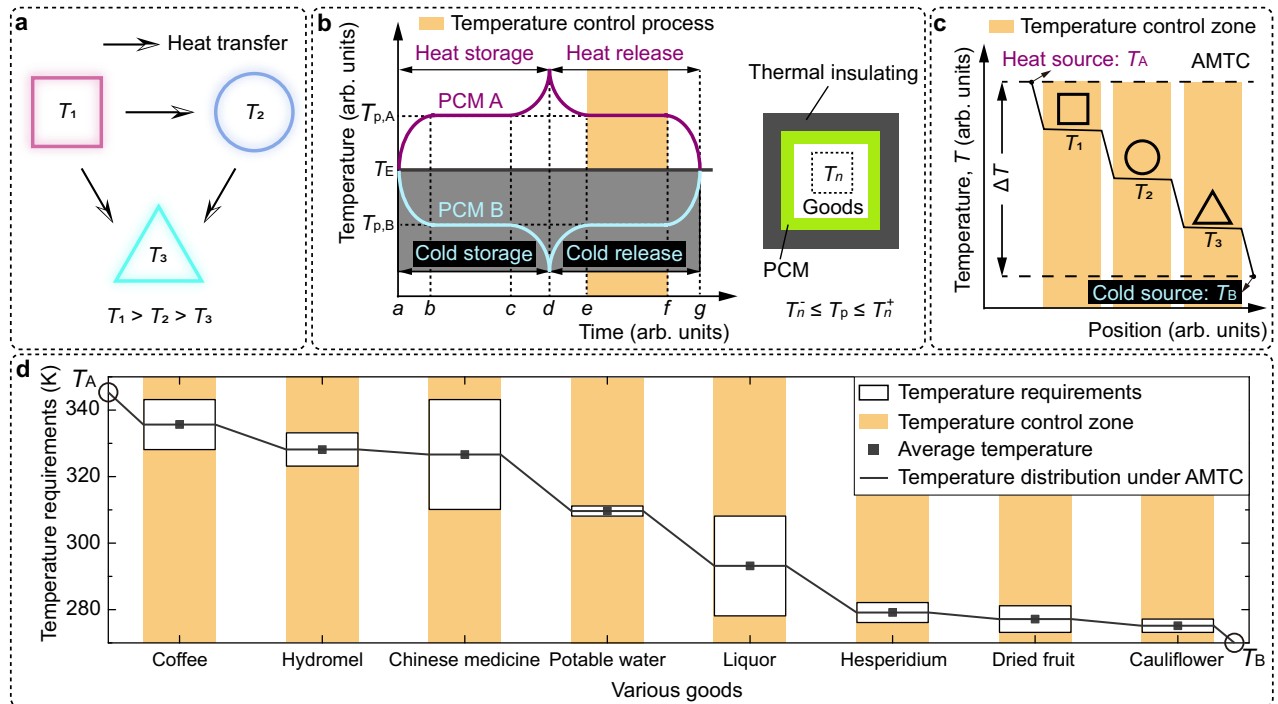

**Fig. 1 | Adaptive multi-temperature control (AMTC). a** Temperature interaction among goods with varying temperature requirements. The square, circle, and triangle symbolize goods with average temperature requirements of $T_1$, $T_2$, and $T_3$, respectively. The temperature requirement ranges for these goods are given by $\left[T_n^-, T_n^+\right]$, where $n = 1,2,3\cdots$ and $T_n = (T_n^- + T_n^+)/2$. **b** Temperature control principle of phase-change materials (PCMs) and their utilization in logistic transportation[2,4,21,24-27]. For generality, units for temperature and time are marked as arbitrary (arb.). $T_{p,A}(T_{p,B})$ represents the phase transition temperature of PCM A (PCM B). $T_E$ is the environmental temperature. $T_{p,A} > T_E$ and $T_{p,B} < T_E$ denote heat (cold) storage, demarcated by the white and gray backgrounds, respectively.

$[a,b] \cup [c,d]$ indicates a sensible heat/cold storage process. $[b,c]$ represents a latent heat/cold storage process. $[d,e] \cup [f,g]$ signifies a sensible heat/cold release process. $[e,f]$ stands for a latent heat/cold release process for realizing temperature control. Please note, the cold storage (heat release) process is presumed ideal, thus disregarding supercooling. **c** Schematic representation of AMTC achieving efficient temperature control for the three types of goods depicted in **a**. **d** Various goods with multi-temperature requirements (extracted from Supplementary Fig. 1). AMTC creates a temperature distribution for efficient multi-temperature control. Source data are provided as a Source Data file.

aforementioned, the current phase-change technology can effortlessly create a temperature difference during transportation by employing a pair of high-performance PCMs with high and low phase transition temperatures as mobile heat and cold sources. By developing various temperature zones that correspond with the temperature requirements of different goods and situating them in the appropriate zones, adaptive multi-temperature control (AMTC) can be achieved (Fig. 1c, d). Since AMTC leverages temperature interactions and does not impose strict temperature requirements on heat and cold sources, it simultaneously circumvents the two overarching challenges of the existing multi-temperature control technology based on multiple PCMs.

Here, we actualized AMTC between a pair of heat and cold sources by designing the system's thermal/structural parameters under conduction heat transfer. Our approach provides a general framework for achieving efficient multi-temperature control by taking advantage of complex temperature interactions among various objects with differing temperature requirements, as opposed to relying solely on thermal insulation and phase-change temperature control, as is habitually done in existing technology. To demonstrate the practical application of our approach, we utilized the AMTC and liquid-solid phase transition of stearic acid and distilled water to fabricate a multi-temperature maintenance container. It provided temperature maintenance for various simulacra at differing temperatures. This work fused the heat flow regulation concept of thermal metamaterials with the existing phase-change technology, addressing critical societal challenges in reliable goods transportation. Our findings and results have the potential for efficient multi-temperature control in diverse fields, encompassing applied physics, engineering science, and life science, ensuring the reliability of objects with varying temperature requirements.

## Results
### Principle of AMTC
Drawing on the concept in Fig. 1c, d, we construct a two-dimensional multi-temperature control system using conduction heat transfer

(Fig. 2). The system comprises a heat source on the left, a cold source on the right, and adiabatic boundaries at the top and bottom. Four different types of zones exist between the heat and cold sources. Zones $C_{\{i,j\}}$ represent areas for storing goods requiring temperature control, referred to as temperature control zones. To achieve multi-temperature control, we set the heat source's temperature higher and the cold source's temperature lower than the maximum and minimum temperature requirements of all zones, respectively. This operation creates a temperature difference between the left and right ends of the system. To control the temperatures of zones $C_{\{i,j\}}$, we need to design the remaining zones $L_{\{i,j\}}$, $R_{\{i,j\}}$, and S.

To start, we fill zones S with thermal insulation materials to prevent temperature interactions along the $Y$ axis. It allows us to focus solely on heat transfer along the $X$ axis. In line with the previous idea, we should make the zones $C_{\{i,j\}}$ at the same temperatures as the temperature requirements of the goods stored in them and with temperature gradient magnitudes close to zero. Following Fourier's law, we finally obtain the relationships between the temperature of each temperature control zone (on row $m$ column $j$) and the thermal/structural parameters under the temperature difference $(T_A \to T_B)$

$$T_{c,\{m,j\}} \approx T_A - \frac{T_A - T_B}{\sum_{j^\#=1}^{n}\sum_{\epsilon=1,r} d_{\epsilon,\{m,j^\#\}}/\kappa_{\epsilon,\{m,j^\#\}}}\left(-\frac{d_{r,\{m,j\}}}{\kappa_{r,\{m,j\}}} + \sum_{j^*=1}^{j}\sum_{\epsilon=1,r}\frac{d_{\epsilon,\{m,j^*\}}}{\kappa_{\epsilon,\{m,j^*\}}}\right),$$
$$(1)$$

where $T_A (T_B)$ is the temperature of the heat (cold) source; $j^\# = \{1,2,3\cdots n\}$, $j^* = \{1,2,3\cdots j\}$; $\epsilon$ is a notation for zone $E$ ($E = \{L,C,R\}$); $\epsilon = l$, $\epsilon = c$, and $\epsilon = r$ denote $E = L$, $E = C$, and $E = R$, respectively; $\kappa_{\epsilon,\{m,j\}}$ ($\kappa_{\epsilon,\{m,j^\#\}}, \kappa_{\epsilon,\{m,j^*\}}$) is the thermal conductivity of $E_{\{m,j\}}$ ($E_{\{m,j^\#\}}, E_{\{m,j^*\}}$); $d_{\epsilon,\{m,j\}}$ ($d_{\epsilon,\{m,j^\#\}}, d_{\epsilon,\{m,j^*\}}$) is the length of $E_{\{m,j\}}$ ($E_{\{m,j^\#\}}, E_{\{m,j^*\}}$) traversed by heat flow (Methods, Supplementary Notes 1–3). Equation (1) indicates that designing the thicknesses (length along the $X$-axis) and thermal conductivities of zones $L_{\{i,j\}}$ and $R_{\{i,j\}}$ can achieve our aim (Supplementary Notes 4, 5, Supplementary Table 1).

Thanks to the universality of AMTC, various multi-temperature control systems can be created to satisfy the varying temperature requirements of goods (Fig. 3). It is important to note that package structures are used to protect these goods. On the characteristic lines, the temperature distributions take on a terraced shape, which is the same as our inspiration in Fig. 1c, d. When applying this approach in real-world settings, it is necessary to ensure that the length and width of the system are reasonable. Since the design method for each row is independent of other rows due to the good thermal insulation of zones S (Fig. 2), we can develop more temperature control zones ($C_{\{i,j\}}$) along the $Y$ axis for storing more goods (Supplementary Note 6, Supplementary Fig. 2).

### Application of AMTC: Multi-temperature maintenance
We simulated a multi-temperature maintenance process to prepare for the transporting of different goods (Fig. 4a, Methods). For simplicity, we assumed nine types of goods required transportation. We used a normalized temperature expression to demonstrate the effectiveness of the multi-temperature maintenance approach (Methods). We began by preheating/precooling the goods to their required temperatures of 0.1, 0.2,..., 0.9. Next, we placed each good into the corresponding temperature control zone within the package structure. To simplify the simulation, we assumed that all goods were square and filled the storage space completely. The transient simulation showed that the AMTC approach effectively maintained the temperatures of the goods after a brief disturbance (Fig. 4b, case 1). In comparison, we replaced the thermal conductivities of the marked zones in the AMTC approach with low thermal conductivity (Fig. 4c, from case 1 to cases 2–4). It is observed that irrespective of how the thermal conductivities were

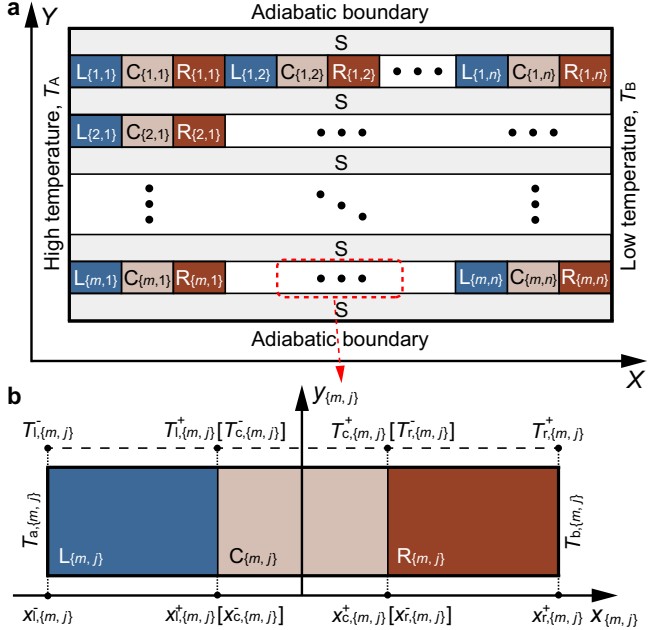

**Fig. 2 | Multi-temperature control system. a** A global system employing conduction heat transfer. Zones $L_{\{i,j\}}$ and $R_{\{i,j\}}$ ($i=1,2,3\cdots,m; j=1,2,3\cdots,n$) are established to manage the temperature gradient from the high-temperature boundary $T_A$ to the low-temperature boundary $T_B$. Zones $C_{\{i,j\}}$ are configured for storing various goods that have different temperature requirements. Zones S are designated to mitigate heat transfer between rows. **b** A local system extracted from row $m$ and column $j$.

reduced, the goods could not achieve the feature of the required multi-temperature maintenance (Fig. 4b, cases 2–4).

We explain the aforementioned phenomenon via the transient net heat flux ($q_N$) of each temperature control zone (Fig. 4d, Supplementary Fig. 4, Supplementary Note 7). For AMTC, there exist two stages. In the first stage, significant net heat flux in each temperature control zone forms to balance the initial temperature difference between the good and its surrounding in a short time. When the goods reach their required temperatures, it transitions to the second stage. Each temperature control zone's transient net heat flux nears zero, exhibiting a multi-temperature maintenance state. In this stage, the temperature interactions among the goods at different temperatures maintain a dynamic balance under the normalized temperature difference ($\Psi_A \rightarrow \Psi_B$). The goods stored in the temperature control zones achieve multi-temperature maintenance. However, without the elaborate design in AMTC and merely reducing the marked zones' thermal conductivities, the transient net heat flux of each temperature control zone is chaotic. This damages the multi-temperature maintenance performance (Fig. 4d, cases 2–4). The comparison shows that the elaborate design of the system's thermal/structural parameters of the AMTC approach is key to achieving efficient multi-temperature maintenance performance whilst taking advantage of the temperature interactions. To further explore the multi-temperature control capability of AMTC, we performed a detailed study on its steady and transient characteristics (Supplementary Notes 8–10). The precise and quick regulation characteristics suggest that AMTC can provide reliable multi-temperature control for various goods with different thermal properties, phases, proportions, shapes, and numbers (Supplementary Figs. 5–13).

## Multi-temperature maintenance container based on AMTC

Based on the aforementioned results, let us fabricate a multi-temperature maintenance container for practical applications (Supplementary Fig. 14). We consider various challenges and their solutions:

- Mobile heat and cold sources: In practical transportation, it is necessary to supply heat and cold to the multi-temperature control system continuously. Therefore, we should prepare a pair of mobile heat and cold sources that are simple, easy to install, and low-cost. As mentioned above, the phase-change technology, which utilizes the latent heat of the liquid-solid PCM, can solve this problem (Fig. 5a). Storing heat (cold) energy in the PCM in advance. When used, the PCM releases heat (cold) energy and maintains an approximately constant temperature (phase transition temperature, $T_p$) during phase transition, acting as a mobile heat (cold) source. After considering the actual temperature requirements of transported goods and the all-around performance of PCMs, we select stearic acid ($T_p \approx 340.15\,K$) and distilled water ($T_p \approx 273.15\,K$) to prepare the mobile heat and cold sources, respectively[59] (Methods, Supplementary Table 2, Supplementary Figs. 15, 16a). For usage, the stearic acid and distilled water must be melted and solidified in advance, respectively. Then, the mobile heat and cold sources can be placed on both sides of the system (Supplementary Fig. 16b) where the environmental temperature ($T_E \approx 294.15\,K$) is between the phase transition temperatures of the stearic acid and the distilled water. Next, the mobile heat (cold) source will release (adsorb) heat, thanks to the latent heat during liquid-solid phase transitions. It allows the mobile heat and cold sources to form a pair of high- and low-temperature boundaries on the left and right sides of the system, which is consistent with the system described in Fig. 2.

- Actual three-dimensional (3-D) multi-temperature control system: To use the prior two-dimensional (2-D) system for practical applications, we need to fill the marked zones with actual materials that produce thermal conductivities equivalent to the theoretical values. Additionally, we aim to minimize the sizes of zones $L_{\{i,j\}}$, $R_{\{i,j\}}$ and S to maximize the storage space (Fig. 2). As a presentation for practical applications, we consider the applicable requirements

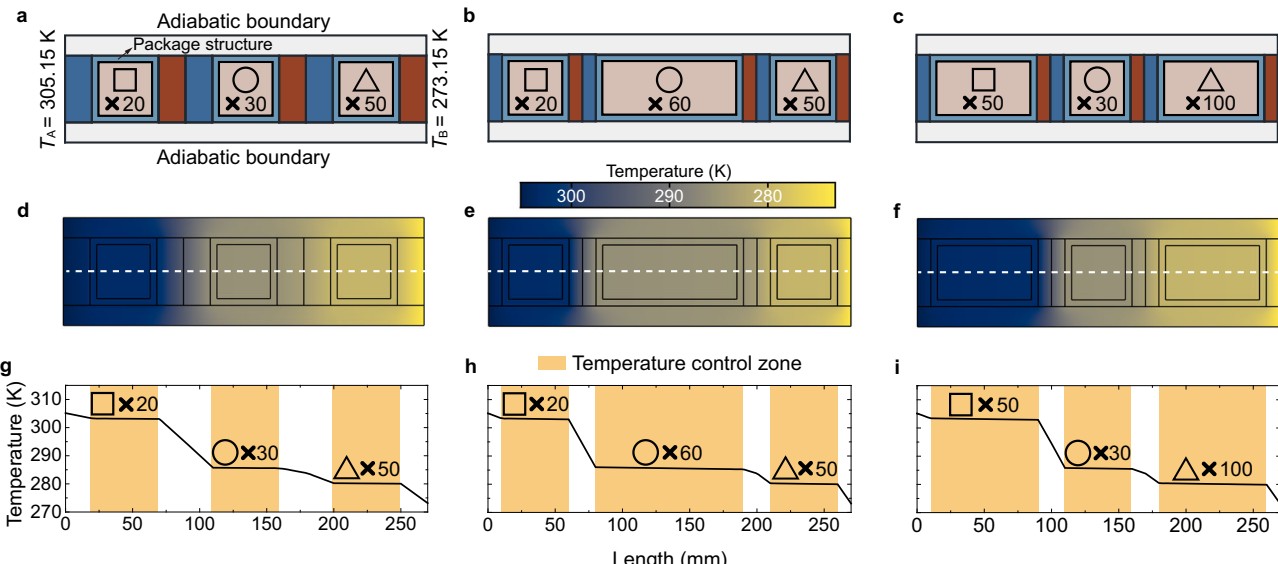

**Fig. 3 | Multi-temperature control capability. a–c** Geometry of different multi-temperature control systems for various goods demands. The temperature requisites for the square ($T_1$), the circle ($T_2$), and the triangle ($T_3$) were set as $T_1 = 303.15\,K$, $T_2 = 285.65\,K$, and $T_3 = 280.15\,K$, corresponding to the actual goods depicted in Supplementary Fig. 1. The high- and low-temperature boundaries are derived from the phase-change temperature of sodium sulfate decahydrate[20] (assuming no undercooling) and water, respectively. To accommodate real-world applications,

we incorporated packaging structures within the temperature control zone to safeguard the goods. For more details, refer to Supplementary Figs. 2b2 and 3. **d–f** Temperature distribution of the multi-temperature control systems under steady state, wherein **d–f** correspond to **a–c**, respectively. The white dashed lines represent characteristic lines utilized for extracting temperature data. **g–i** Temperature distribution along the characteristic lines, where **g–i** are aligned with **d–f**, respectively. Source data are provided as a Source Data file.

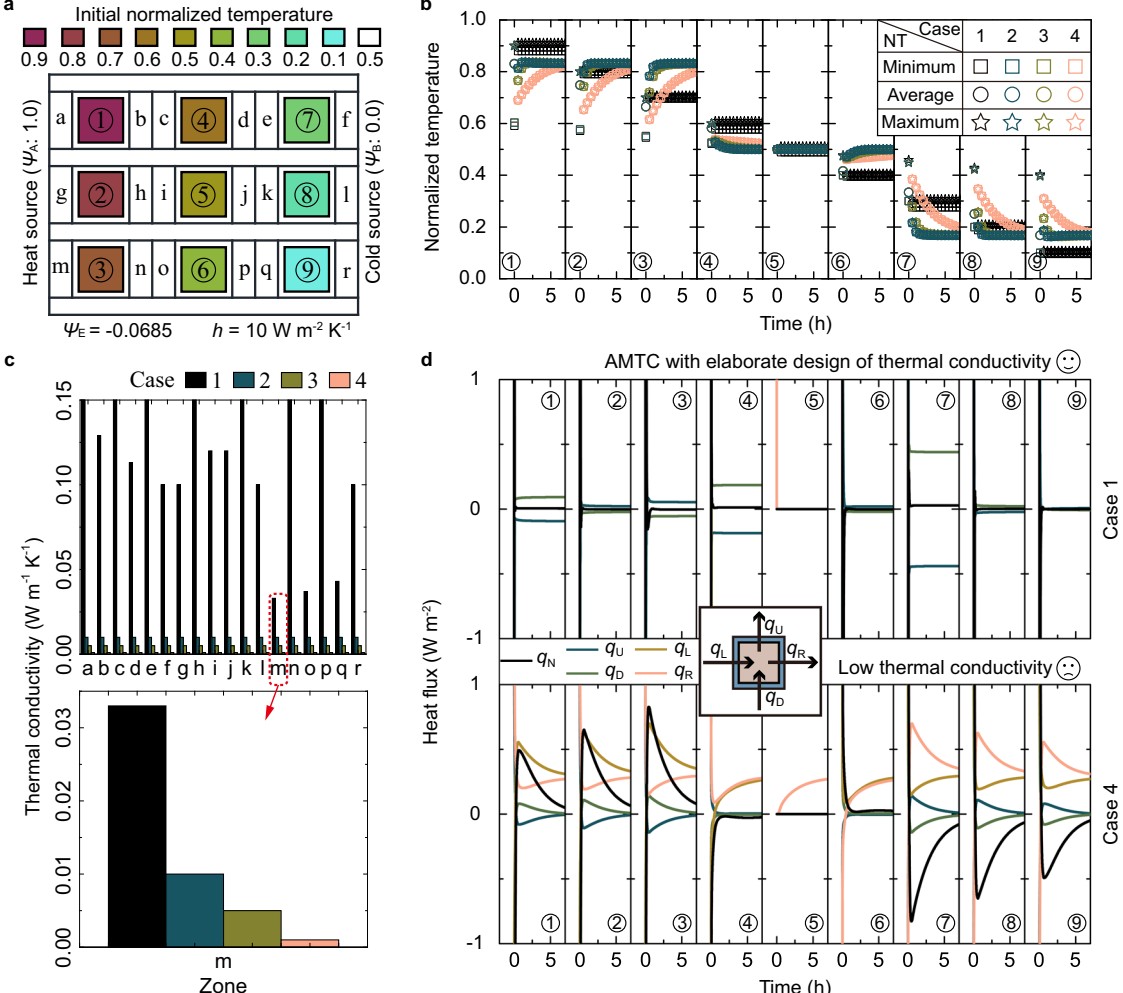

**Fig. 4 | Multi-temperature maintenance capability. a** Schematic diagram of the multi-temperature maintenance. The normalized environmental temperature ($\Psi_E$) was −0.0685. The convective heat transfer coefficient ($h$) of the upper and lower boundaries was 10 W m⁻² K⁻¹. We show the initial normalized temperature of each part in the color bar. Symbols "a–r" indicate regions where thermal conductivities require design. **b** Minimum/average/maximum normalized temperature (NT) plotted against time for cases 1–4. **c** Thermal conductivities for the zones marked in **a**. Four cases are presented: 1, 2, 3, and 4. In case 1, all thermal conductivities were designed by AMTC. For cases 2, 3, and 4, thermal conductivities were replaced with values of 0.01 W m⁻¹ K⁻¹, 0.005 W m⁻¹ K⁻¹, and 0.001 W m⁻¹ K⁻¹, respectively. For clarity, the range of thermal conductivities is set to 0–0.15 W m⁻¹ K⁻¹. An inset offers a comparative view of thermal conductivity values across the four cases, using the "m" zone as an example. **d** Heat flux plotted against time for each temperature control zone in cases 1 and 4. An inset displays the schematic of heat flux across temperature control zones from the left ($q_L$), right ($q_R$), up ($q_U$), and down ($q_D$) sides. For case 1, $q_L$ and $q_R$ exceed the given range. Further details can be found in Supplementary Fig. 4. Source data are provided as a Source Data file.

(Fig. 1d) and then set the required temperature ($T$) of each temperature control zone ($\gamma$) to

$$\left[T_{\gamma,\mathrm{re}}\right]_{3\times3} = \begin{bmatrix} T_① & T_④ & T_⑦ \\ T_② & T_⑤ & T_⑧ \\ T_③ & T_⑥ & T_⑨ \end{bmatrix} = \begin{bmatrix} 330.10 & 313.35 & 289.90 \\ 326.75 & 306.65 & 286.55 \\ 323.40 & 299.95 & 283.20 \end{bmatrix} \mathrm{K} \quad (2)$$

for transporting goods with similar temperatures. In line with the above requirements, we designed an actual 2-D multi-temperature control system (Supplementary Note 11, Supplementary Figs. 17–20, Supplementary Tables 3, 4). Since Supplementary Note 9.5 indicates that the design method of a 2-D system is also valid for 3-D, we can directly extend the 2-D system to 3-D (Fig. 5a, Supplementary Fig. 21). For storage requirements, the system's height should be not less than those of the goods.

- Assembly and thermal insulation of multi-temperature control system: The multi-temperature control is realized by conduction heat transfer, so it is crucial to connect all components tightly

(Supplementary Fig. 16b). The assembly method should be as simple as possible. In addition, to eliminate the influence of the outside environment on the multi-temperature control performance, the system's outer boundary should maintain good thermal insulation. Considering the above factors, we put the multi-temperature control system into a commercial thermal insulating container with a suitable size. Finally, we fabricated a multi-temperature maintenance container (Fig. 5b, c; Methods). As mentioned above, it can transport goods with temperature requirements close to the elements in Eq. (2). Supplementary Note 12 compares the multi-temperature maintenance container with the commercial and phase-change thermal insulating containers in existing logistics. It indicates that the rational utilization of temperature interactions in the multi-temperature control system mentioned above is different from the thermal insulating operation in standard technologies, which might be beneficial for efficient multi-temperature control (Supplementary Fig. 14).

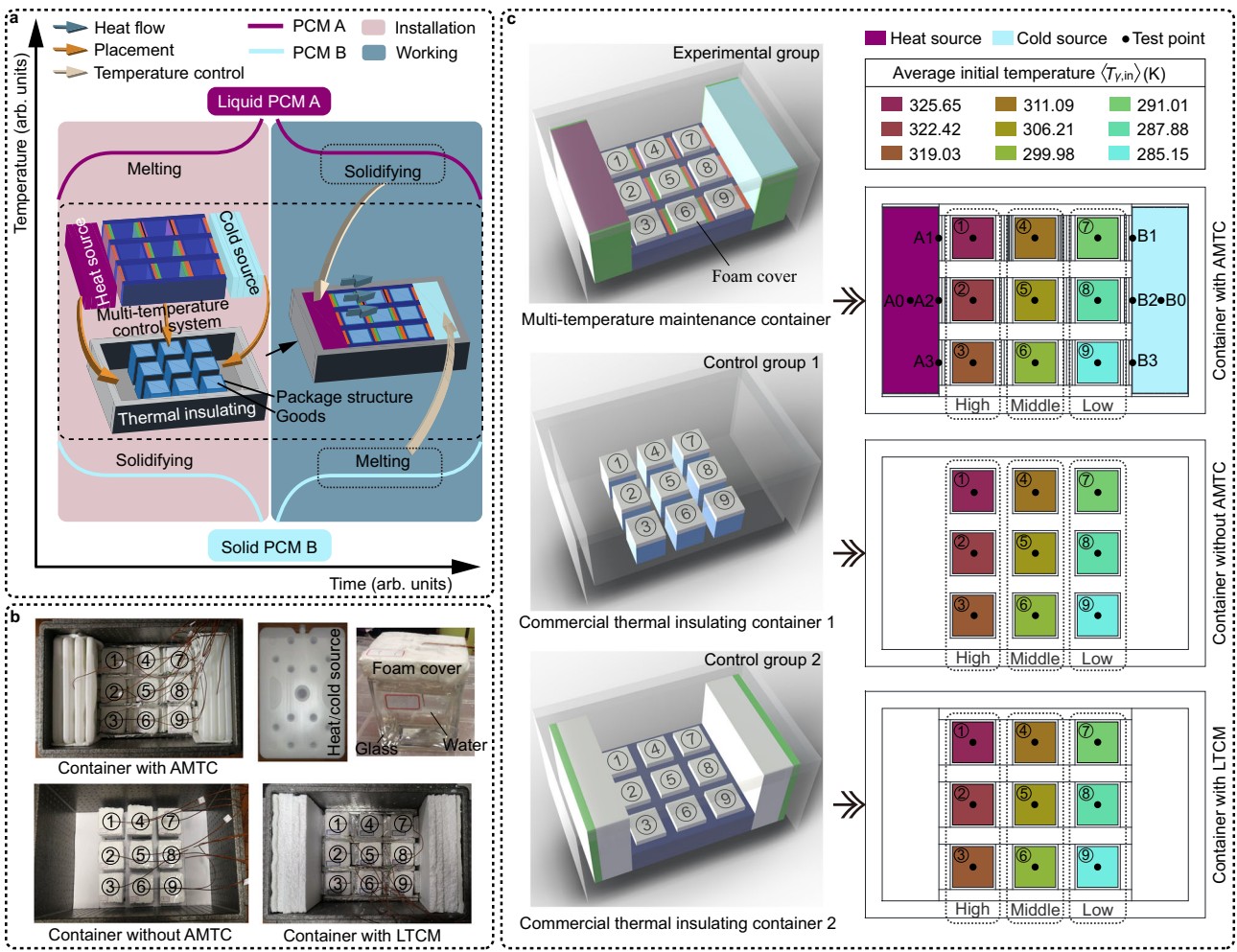

**Fig. 5 | Multi-temperature maintenance container. a** Schematic diagram of a multi-temperature maintenance container. To ensure generality, the units used for temperature and time are arbitrary (arb.). **b**, **c** Experimental setup (**b**) and schematic diagram (**c**) for the multi-temperature maintenance container (container with AMTC (adaptive multi-temperature control)), commercial thermal insulating container 1 (container without AMTC), and commercial thermal insulating container 2 (container with LTCM (low thermal conductivity material)), Supplementary Fig. 22). The test points A1-A3 and B1-B3 indicate the temperature boundaries provided by the mobile heat and cold sources, respectively. The test points A0 and B0 reflect the temperatures of heat and cold sources, respectively. The squares in different colors denote the average initial temperatures of the simulacrum in each temperature control zone (Methods).

## Performance of the multi-temperature maintenance container

To assess the performance of our multi-temperature maintenance container (Fig. 5b, c; experimental group: the container with AMTC), we should obtain the temperature-time curves of the goods stored in the container (Supplementary Note 13, Supplementary Fig. 23). We prepared nine volumes of water, all of equal measure but of varying initial temperatures, to simulate the different temperature requirements of goods (Methods, Supplementary Figs. 24, 25, Supplementary Table 5). These were then placed within their corresponding temperature control zones. After the container was sealed, we monitored the real-time changes in temperature. The performance of the AMTC is considered superior if there are smaller temperature changes.

For comparative purposes, we set two control groups. The first group was a container without the AMTC, and the second was a container that had the AMTC replaced with a low thermal conductivity material (LTCM). Both these scenarios are common practice in commercially available thermal insulating containers. We examined the temperature variations of the simulated items within the containers both with and without the AMTC and with the LTCM. This comparison served to illustrate the influence of the AMTC on maintaining multiple temperatures. For ease of analysis, we divided the zones (①, ②, ③)/(④, ⑤,

⑥)/(⑦, ⑧, ⑨) into a high-/middle-/low-temperature region (Fig. 5c) based on their preset temperatures [Eq. (2)].

Following the multi-temperature maintenance test, we were able to observe the temperature changes of the simulacra stored in the containers (Fig. 6a, b). In the container without AMTC, the items in the high-temperature region cooled rapidly, while those in the low-temperature region warmed up and surpassed the surrounding temperature within approximately one hour. Despite the temperature variations being somewhat controlled in the container with the LTCM, achieving thorough multi-temperature control was not feasible. However, within the container equipped with the AMTC, temperature variations of the simulacra were further minimized under the constant temperature difference provided by mobile heat and cold sources (Fig. 6c, d).

To delve deeper into the workings of multi-temperature maintenance, we performed finite element simulations to visually depict the temperature distribution within the containers at distinct times (Fig. 7a). Our heat flux analysis for each temperature control zone revealed that the irregular temperature interactions among the simulacra at different temperatures in the container without AMTC disrupted multi-temperature maintenance, particularly in the high- and low- temperature regions (Fig. 7b). We quantitively assessed the multi-

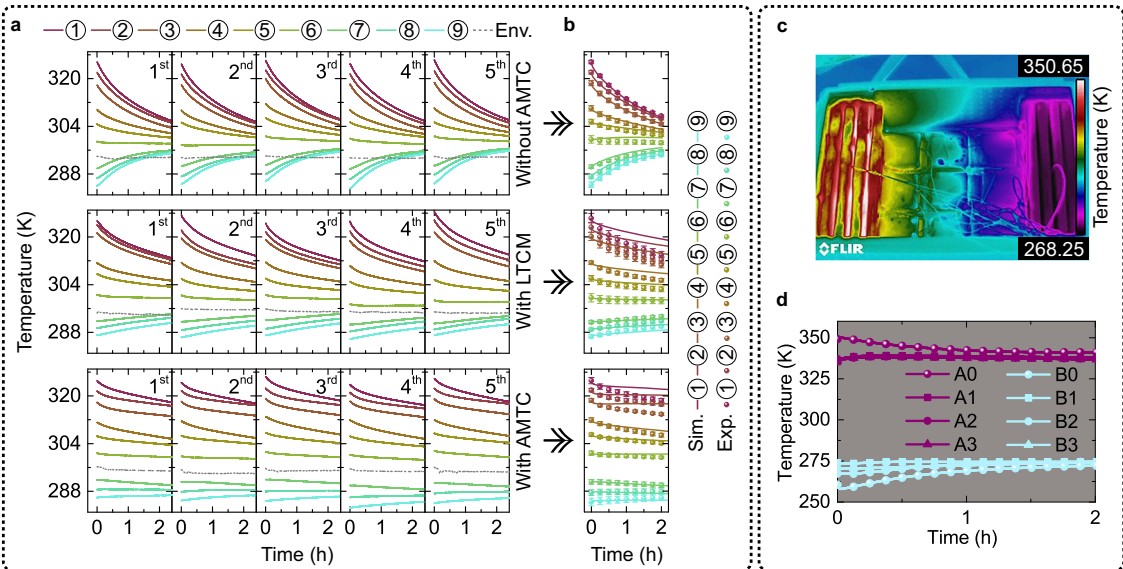

**Fig. 6 | Multi-temperature maintenance test. a** Experimental results of the simulacra's temperature over time in the container without AMTC (adaptive multi-temperature control), with LTCM (low thermal conductivity material), and with AMTC, under five independent tests, with indoor environment temperature (Env.) as reference. **b** Comparison of simulation (Sim.) and experiment (Exp.) results on the temperature-time curves of the simulacra stored in the containers. **c** Infrared top view of the multi-temperature maintenance container upon opening the lid. **d** Experimental results of temperatures on test points A0–A3 and B0–B3 over time. We employ a gray background to improve color resolution. Source data are provided as a Source Data file.

temperature maintenance performance of the container by calculating the average temperature variation rate $\langle \eta_{\gamma,\tau,u} \rangle$ (Fig. 7c). After 2 hours, the temperature variation magnitude $|\langle \eta_{\gamma,\tau,u} \rangle|$ of simulacra stored in the container with AMTC was merely 0.14–2.05% (Supplementary Fig. 26). When comparing the multi-temperature maintenance performance of the container without AMTC, with LTCM, and with AMTC, we found that the more the temperature interactions disrupt the multi-temperature maintenance performance, the great the effect of AMTC. Specifically, in the high- (low-) temperature regions, $|\langle \eta_{\gamma,\tau,u} \rangle|$ in the container with AMTC was 53.3–69.2% (78.0–94.7%) lower than those without AMTC and 34.7–56.3% (33.0–85.9%) lower than those with LTCM (Fig. 7d, e). The improvement in the multi-temperature maintenance performance made possible by AMTC highlights its potential for reliable transportation of essential items.

## Discussion

In summary, we developed an AMTC approach enabled by PCMs for transporting essential items. Our methodology focuses on an elaborate design of the system's thermal and structural parameters, utilizing both high- and low-temperature PCMs, instead of conventional strategies with one PCM per zone and thermal insulation. By leveraging temperature interactions among the transported items with varying temperature needs, AMTC has proven to be effective.

To demonstrate the practicality of our approach, we constructed a multi-temperature maintenance container based on AMTC, using stearic acid and distilled water for the liquid-solid phase transitions. This container effectively maintains the temperatures of nine simulacra representing various goods, with minimal temperature variations of only 0.14–2.05% after 2 hours. Compared to a container without AMTC and with LTCM (low thermal conductivity materials), our multi-temperature maintenance container reduces temperature variations by up to 94.7% and 85.9%, respectively, highlighting the effectiveness of AMTC in ensuring reliable transportation of goods.

Moreover, the universality of AMTC allows for convenient adjustment of the sizes, quantities, and temperatures of the temperature control zones within the multi-temperature maintenance container to meet users' specific requirements. As high-performance composite PCMs continue to advance, the energy densities and

thermal properties of the mobile heat and cold sources can be further enhanced, leading to improved multi-temperature maintenance performance[60–63].

The development of stable supercooling and phase-change hysteresis technologies, which can provide stable temperature boundaries for AMTC, can expand the range of PCM candidates for mobile heat sources and increase energy storage convenience with the seasonal storage characteristics[35–38,62,64–68]. We also considered critical application problems such as size enlargement, wrongly placed, and thermal inertia of the container with AMTC (Supplementary Note 14, Supplementary Figs. 27–30, Supplementary Table 6). The stable multi-temperature maintenance performance of the container in the case of proportional amplification, the protection characteristics of the non-misplaced objects in the case of wrong placement, and the thermal inertia gain consistent with the traditional technology further ensure the application value of AMTC.

By combining the concept of heat flow regulation in thermal metamaterials with phase-change technologies, our approach addresses critical social challenges associated with reliable goods transportation. Unlike advanced temperature maintenance methods that primarily focus on precise heat flow regulation for a single zone[41], our approach allows for the creation of multiple zones with adjustable temperatures, offering practicality, portability, and storage-oriented features.

In comparison to other temperature maintenance methods that use phase-change technologies to construct multi-temperature insulating boxes for cold chain logistics using composite PCMs with suitable phase-change temperatures[2,4], our approach reduces the demand for heat and cold sources while expanding the number of temperature control zones. This simplifies the application process and initial investment. Consequently, our work combines the fundamental advantages of physics with the practical value of engineering science, resulting in easy implementation and an increased number of temperature control zones.

From a biological standpoint, the mechanism of AMTC in ensuring the quality of essential goods lies in its ability to control the activity of internal microbes. For instance, as depicted in Supplementary Fig. 1, AMTC simultaneously inhibits and preserves microbial activities in

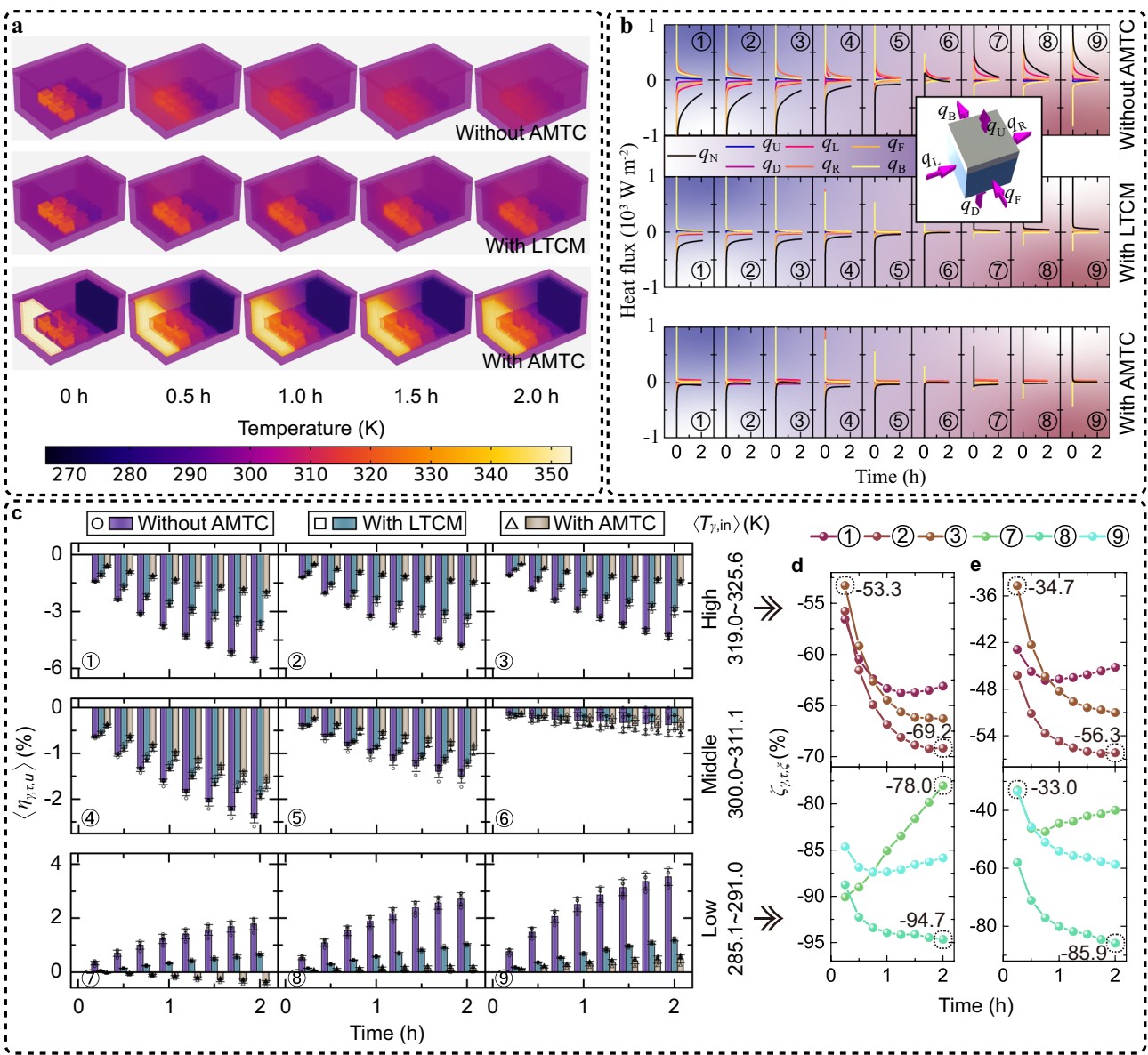

**Fig. 7 | Effect of adaptive multi-temperature control (AMTC). a** Simulation of the temperature distributions inside the containers without AMTC, with LTCM (low thermal conductivity material), and with AMTC. Images within the same group are distinguished with a gray background. **b** Heat flux analysis for each temperature control zone (Supplementary Note 7). We utilize a gradient background to sharpen the distinction in various color plots. **c** Temperature variation rates ($\eta_{\gamma,\tau,u}$) of the simulacra stored in the containers at characteristic times. All $\eta_{\gamma,\tau,u}$ were averaged from five test results, while the error bars represent the experimental standard deviation $\sigma_{\gamma,\tau,u}$ (Methods). The five results for the case without AMTC, with LTCM, and with AMTC are represented by a circle, square, and triangle, respectively, to illustrate the distribution of the underlying data. **d**, **e** Change in simulacra's temperature variation magnitudes $\zeta_{\gamma,\tau,\xi}$ ($\xi=1,2$) in the high- and low-temperature regions at the characteristic times correspond to **c** (Methods). The multi-temperature maintenance performance improvement moving from the case without AMTC (with LTCM) to that with AMTC is represented by $\zeta_{\gamma,\tau,1}$ ($\zeta_{\gamma,\tau,2}$), as depicted in **d** (**e**). Source data are provided as a Source Data file.

foods and vaccines, respectively. The temperature plays a crucial role in controlling microbial activity[69], and AMTC demonstrates its capability to achieve multi-level temperature-controlled microbial activities, even under complex temperature interactions, thereby addressing the social challenges associated with the reliable transportation of goods. Consequently, this approach offers an efficient means to control the diverse activity requirements of microbes. As a result, our findings and results present a potential for achieving efficient multi-temperature control across various fields, including applied physics, engineering science, life science, and beyond, ensuring the reliability of objects with diverse temperature requirements.

In terms of future developments, the adiabatic assumptions factoring in individual boundaries may be influenced by the surrounding environment, resulting in some level of heat dissipation. Viewed three-dimensionally, three pairs of heat and cold sources can be set up between opposing sides, akin to the utilization of PCM in cold chain logistics[19,21,70]. Using the AMTC approach as a basis, the temperature within each zone can be modulated, thereby achieving multi-temperature joint control that fully insulates from external temperature variations. Additionally, a preliminary model analysis has been conducted to elucidate the relationship between the effective multi-temperature maintenance time, the phase-change temperature of PCMs, and the system's thermal and structural parameters (Supplementary Note 15). A more sensitive analysis inclusive of the physicochemical properties of PCMs and the properties of the goods, with a specific focus on the multi-temperature control process, could potentially enhance the multi-temperature maintenance container. This could also optimize the strategy for selecting actual composite

PCMs with suitable phase-change temperatures and designing systems with superior thermal and structural parameters for storing goods in appropriate quantities.

## Methods
### General theory for AMTC
In accordance with Fourier's law, heat flux along the $X$ axis is articulated as

$$q = -\kappa \frac{\mathrm{d}T}{\mathrm{d}X}, \tag{3}$$

with $\kappa$ denoting thermal conductivity, $T$ representing temperature, and $X$ signifying position. We select a local system (Fig. 2b) situated on row $m$, column $j$, with $T_{\mathrm{a},\{m,j\}}$ as the temperature on the left side and $T_{\mathrm{b},\{m,j\}}$ on the right. Assuming $\kappa$ does not depend on $T$, we apply Eq. (3) to derive the temperature distribution in $C_{\{m,j\}}$ from the viewpoint of the local system as follows:

$$T_{\mathrm{c},\{m,j\}} = q_m \left( \frac{d_{\mathrm{r},\{m,j\}}}{\kappa_{\mathrm{r},\{m,j\}}} + \frac{d_{\mathrm{c},\{m,j\}}/2 - x_{\{m,j\}}}{\kappa_{\mathrm{c},\{m,j\}}} \right) + T_{\mathrm{b},\{m,j\}}. \tag{4}$$

In Eq. (4), $q_m$ refers to the heat flux that traverses $C_{\{m,j\}}$ along the $X$ axis on row $m$; $d_{\mathrm{r},\{m,j\}}$ and $d_{\mathrm{c},\{m,j\}}$ depict the lengths of $R_{\{m,j\}}$ and $C_{\{m,j\}}$ traversed by the heat flow, respectively; $\kappa_{\mathrm{r},\{m,j\}}$ and $\kappa_{\mathrm{c},\{m,j\}}$ are thermal conductivities of $R_{\{m,j\}}$ and $C_{\{m,j\}}$, respectively; $x_{\{m,j\}}$ denotes the position in the local system (See detailed derivation in Supplementary Note 1.1). Note that the temperature control zone comprises a package structure and a storage space, treated here as a single entity for the sake of simplicity (Supplementary Note 2). Equation (4) elucidates a temperature control mechanism for goods stored in $C_{\{m,j\}}$ requiring a specific temperature, $T_{\mathrm{c},\{m,j\},\mathrm{re}}$. The method involves adjusting the system's thermal and structural parameters to modulate the heat flux ($q_m$) between the goods and their surroundings until the goods' temperature ($T_{\mathrm{o},\{m,j\}}$) reaches the necessary value ($T_{\mathrm{o},\{m,j\},\mathrm{re}}$). However, when viewed from the perspective of the global system, the temperature interactions among local systems render multi-temperature control more intricate and challenging. Specifically, both $q_m$ and $T_{\mathrm{b},\{m,j\}}$ in Eq. (4) depend on the global system's thermal/structural parameters and the temperatures of heat and cold sources ($T_A \& T_B$)

$$q_m = \frac{T_A - T_B}{\sum_{j^\#=1}^{n} \sum_{\epsilon=1,\mathrm{c},\mathrm{r}} d_{\epsilon,\{m,j^\#\}} / \kappa_{\epsilon,\{m,j^\#\}}}, \tag{5a}$$

$$T_{\mathrm{b},\{m,j\}} = T_A - (T_A - T_B) \sum_{j'=1}^{j} \frac{\sum_{\epsilon=1,\mathrm{c},\mathrm{r}} d_{\epsilon,\{m,j'\}} / \kappa_{\epsilon,\{m,j'\}}}{\sum_{j^\#=1}^{n} \sum_{\epsilon=1,\mathrm{c},\mathrm{r}} d_{\epsilon,\{m,j^\#\}} / \kappa_{\epsilon,\{m,j^\#\}}}. \tag{5b}$$

Equation (5a) is obtained according to the equivalent thermal resistance method. See the detailed derivation for Eq. (5b) in Supplementary Note 1.2. To attain multi-temperature control, it is essential to consider the temperature interactions among local systems. By substituting Eq. (5a) and Eq. (5b) into Eq. (4), we obtained a temperature expression for $C_{\{m,j\}}$ from the global system perspective

$$T_{\mathrm{c},\{m,j\}} = T_A - \frac{T_A - T_B}{\sum_{j^\#=1}^{n} \sum_{\epsilon=1,\mathrm{c},\mathrm{r}} d_{\epsilon,\{m,j^\#\}} / \kappa_{\epsilon,\{m,j^\#\}}} \left( -\frac{d_{\mathrm{r},\{m,j\}}}{\kappa_{\mathrm{r},\{m,j\}}} \right.$$
$$\left. -\frac{d_{\mathrm{c},\{m,j\}}/2 - x_{\{m,j\}}}{\kappa_{\mathrm{c},\{m,j\}}} + \sum_{j^*=1}^{j} \sum_{\epsilon=1,\mathrm{c},\mathrm{r}} \frac{d_{\epsilon,\{m,j^*\}}}{k_{\epsilon,\{m,j^*\}}} \right). \tag{6}$$

We can thus control the temperature of the goods by adjusting the system's thermal/structural parameters. Furthermore, we set

$\kappa_{\mathrm{l},\{m,j^\#\}} \ll \kappa_{\mathrm{c},\{m,j\}}$ and $\kappa_{\mathrm{r},\{m,j^\#\}} \ll \kappa_{\mathrm{c},\{m,j\}}$. Then, the temperature in $C_{\{m,j\}}$ is uniform ($|\nabla T_{\mathrm{c},\{m,j\}}| \to 0$). See detailed derivation in Supplementary Note 1.3). Consequently, we finally determine the temperature of the goods in $C_{\{m,j\}}$ ($T_{\mathrm{o},\{m,j\}}$), as indicated in Eq. (1). See detailed derivation in Supplementary Note 1.4.

### Normalized expression and post-processing of the simulation results
In the "Application of AMTC: Multi-temperature maintenance" and Supplementary Note 8, we employ the normalized expression to exhibit the characteristics of the 2-D multi-temperature control system. We now introduce the detailed method. We define the relationship between temperature $T$ and normalized temperature $\Psi$ as

$$T = \Psi (T_A - T_B) + T_B. \tag{7}$$

Subsequently, we present the normalized expression of the good's temperature requirement, the heat source temperature, and the cold source temperature as

$$
\begin{aligned}
\Psi_{\mathrm{o},\{m,j\},\mathrm{re}} &= \frac{T_{\mathrm{o},\{m,j\},\mathrm{re}} - T_B}{T_A - T_B}, \\
\Psi_A &= \frac{T_A - T_B}{T_A - T_B} = 1.0, \\
\Psi_B &= \frac{T_B - T_B}{T_A - T_B} = 0.0,
\end{aligned}
\tag{8}
$$

where $T_{\mathrm{o},\{m,j\},\mathrm{re}}$ represents the required temperature of the goods. In consideration of Eq. (1), we have

$$
\begin{aligned}
\Psi_{\mathrm{o},\{m,j\}} &\approx 1 - \frac{1-0}{\sum_{j^\#=1}^{n} \sum_{\epsilon=1,\mathrm{r}} d_{\epsilon,\{m,j^\#\}} / \kappa_{\epsilon,\{m,j^\#\}}} \left( -\frac{d_{\mathrm{r},\{m,j\}}}{\kappa_{\mathrm{r},\{m,j\}}} + \sum_{j^*=1}^{j} \sum_{\epsilon=1,\mathrm{r}} \frac{d_{\epsilon,\{m,j^*\}}}{k_{\epsilon,\{m,j^*\}}} \right) \\
&\approx \Psi_A - \frac{\Psi_A - \Psi_B}{\sum_{j^\#=1}^{n} \sum_{\epsilon=1,\mathrm{r}} d_{\epsilon,\{m,j^\#\}} / \kappa_{\epsilon,\{m,j^\#\}}} \left( -\frac{d_{\mathrm{r},\{m,j\}}}{\kappa_{\mathrm{r},\{m,j\}}} + \sum_{j^*=1}^{j} \sum_{\epsilon=1,\mathrm{r}} \frac{d_{\epsilon,\{m,j^*\}}}{k_{\epsilon,\{m,j^*\}}} \right).
\end{aligned}
\tag{9}
$$

On comparing Eq. (1) with Eq. (9), it is found that the system's thermal/structural parameters remain unaltered in both equations. Additionally, we hypothesize that there is no internal heat source in the multi-temperature control system. Therefore, the heat conduction differential equation can be expressed as

$$\rho c \frac{\partial T}{\partial t} = \nabla \cdot (\kappa \nabla T), \tag{10}$$

where $\rho$ signifies density, $c$ stands for specific heat capacity, and $t$ denotes time. By substituting Eq. (7) into Eq. (10), we have

$$\rho c \frac{\partial \Psi}{\partial t} = \nabla \cdot (\kappa \nabla \Psi). \tag{11}$$

The formal consistency between Eq. (10) and Eq. (11) ensures the consistency of temperature and normalized temperature expressions, which characterize the temperature control process of the system. Hence, we directly convert the temperature data in the simulation results into normalized temperature data, following Eq. (7).

### Design method of a 2-D multi-temperature control system
We show the overall design idea of the 2-D multi-temperature control system in Supplementary Note 4. We now introduce the fundamental calculating methods of the system's thermal/structural parameters. We use the temperature expression in the calculating parts rather than the normalized temperature expression, to ensure easier comprehension. Equation (1) yields a relationship between $T_{\mathrm{c},\{m,j\}}$ and the

system's thermal/structural parameters [$\kappa_{l,\{m,j\}}$,$\kappa_{r,\{m,j\}}$,$d_{l,\{m,j\}}$ and $d_{r,\{m,j\}}$] and the heat & cold sources ($T_A$ & $T_B$). Considering the temperature difference $T_A \to T_B$, we have $4n$ adjustable thermal/structural parameters to align $T_{c,\{m,j\}}$ with $T_{o,\{m,j\},re}$, the required temperature of the goods. The count of $T_{o,\{m,j\},re}$ is $n$. By pre-setting $d_{l,\{m,j\}}$ and $d_{r,\{m,j\}}$, we reduce the number of adjustable parameters from $4n$ to $2n$. Further pre-setting of $n$ thermal parameters decreases the remaining adjustable parameters from $2n$ to $n$. Consequently, Eq. (1) can be solved, and all the thermal parameters ($\kappa_{l,\{m,j\}}$ and $\kappa_{r,\{m,j\}}$) are obtained. As $m$ is arbitrary, we can acquire $\kappa_{l,\{i,j\}}$ and $\kappa_{r,\{i,j\}}$ ($i = 1,2,3,\cdots,m$) by repeating the above operations along $Y$ axis (Supplementary Note 5). A simplified approach based on normalized temperature expression is presented in Supplementary Note 1.5. Additionally, Supplementary Note 5 enumerates six approaches for computing the system's thermal/structural parameters. All the calculation results have been verified by finite element simulations.

## Finite element simulations

For the sake of convenience, we first present the methods for steady simulation of Supplementary Fig. 5a–f. Supplementary Fig. 5a illustrates the geometry and boundary conditions of the multi-temperature control system. To simplify the discussion, we used the temperature expression for simulation and set $\Psi_A = 1.0$ corresponding to $T_A = 400\,K$, and $\Psi_B = 0.0$ corresponding to $T_B = 300\,K$. The detailed structural parameters of this model are provided in Supplementary Fig. 2b1. For the steady simulation, the thermal parameters ($\kappa_{l,\{i,j\}}$, $\kappa_{r,\{i,j\}}$, $\kappa_{o,\{i,j\}}$, $\kappa_{pac,\{i,j\}}$ and $\kappa_s$; $\kappa_{o,\{i,j\}}$ and $\kappa_{pac,\{i,j\}}$ are the thermal conductivities of the good and the package structure, respectively) are shown in Supplementary Note 4. We performed the steady simulation using COMSOL Multiphysics and obtained the temperature distribution of the system (Supplementary Fig. 5b). To further analyze the goods' temperature in each temperature control zone, we utilized "dot probes" to extract the maximum/average/minimum temperature of the goods (Supplementary Fig. 5c). Additionally, we extracted the temperature distribution along characteristic lines in the first, second, and third rows of the system (Supplementary Fig. 5d). To assess the multi-temperature control performance practically, we removed the goods from the storage space and conducted tests under these conditions. The geometry and boundary conditions of the system in Supplementary Fig. 5e were the same as those in Supplementary Fig. 5a. By setting $\kappa_{o,\{i,j\}} = 0.029\ W\,m^{-1}\,K^{-1}$ and keeping the other parameters unchanged, we obtained the temperature distribution of the system in this scenario (Supplementary Fig. 5f). We compared the cases with filled and unfilled goods (Supplementary Fig. 5a, e) by extracting the characteristic temperature distributions in the first, second, and third rows (Supplementary Fig. 5b, f, and Supplementary Fig. 5d).

Then, we introduce the methods of transient simulations of Supplementary Fig. 5g–j. Supplementary Note 4 has already provided the thermal conductivities of the system. Now, we introduce the remaining parameters for the transient simulation: the density ($\rho_\xi$) and specific heat capacity ($c_\xi$) of the system components $\Xi$. Here, $\xi$ denotes a notation of $\Xi$, $\Xi = \{L_{\{i,j\}}, R_{\{i,j\}}, S\}$; $\xi = l$, $\xi = r$, and $\xi = s$ correspond to $\Xi = L_{\{i,j\}}$, $\Xi = R_{\{i,j\}}$, and $\Xi = S$, respectively. For simplicity, we set $\rho_\xi = \rho_l = \rho_r = \rho_s = 100\ kg\,m^{-3}$ and $c_\xi = c_l = c_r = c_s = 100\ J\,kg^{-1}\,K^{-1}$. The densities of the goods and package structures were set to $100\ kg\,m^{-3}$, and their specific heat capacities were set to $100\ J\,kg^{-1}\,K^{-1}$. We performed the transient simulation and revealed the goods' maximum/average/minimum temperatures versus time using "dot probes" (Supplementary Fig. 5g). To analyze the mechanism of AMTC, we extracted the heat fluxes of each temperature control zones in the left, right, up, and down directions through "probes" (Supplementary Fig. 5h, i). To study the transient characteristics of AMTC further, we regulated the thermal diffusivities ($\alpha_\xi$) of the system components by adjusting $c_\xi$. We defined a characteristic parameter ($\beta$) to represent the

components' thermal diffusivities. For $\beta = 50$, $c_\xi = 2\ J\,kg^{-1}\,K^{-1}$; for $\beta = 10$, $c_\xi = 10\ J\,kg^{-1}\,K^{-1}$; for $\beta = 1$, $c_\xi = 100\ J\,kg^{-1}\,K^{-1}$ (the same as in Supplementary Fig. 5g, serving as a control group); for $\beta = 0.1$, $c_\xi = 1000\ J\,kg^{-1}\,K^{-1}$; for $\beta = 0.02$, $c_\xi = 5000\ J\,kg^{-1}\,K^{-1}$. It indicates that the larger the $\beta$, the larger the $\alpha_\xi$. The transient simulation results for goods' maximum/average/minimum temperatures versus time with different thermal diffusivities are shown in Supplementary Fig. 5j.

Next, we introduce the transient simulation methods of Figs. 3 and 4. This section consists of two parts. The first part deals with steady simulations of AMTC with different good requirements (Fig. 3a–c). The structural parameters of the systems can be found in Supplementary Figs. 2b2 and 3. The left and right boundaries were set to 305.15 K and 273.15 K, respectively. The upper and lower boundaries were adiabatic. For Fig. 3a, $\kappa_{l,\{1,1\}} = 0.036\ W\,m^{-1}\,K^{-1}$, $\kappa_{r,\{1,1\}} = 0.008\ W\,m^{-1}\,K^{-1}$, $\kappa_{l,\{1,2\}} = 0.008\ W\,m^{-1}\,K^{-1}$, $\kappa_{r,\{1,2\}} = 0.040\ W\,m^{-1}\,K^{-1}$, $\kappa_{l,\{1,3\}} = 0.019\ W\,m^{-1}\,K^{-1}$, and $\kappa_{r,\{1,3\}} = 0.010\ W\,m^{-1}\,K^{-1}$. For Fig. 3b, $\kappa_{l,\{1,1\}} = 0.037\ W\,m^{-1}\,K^{-1}$, $\kappa_{r,\{1,1\}} = 0.008\ W\,m^{-1}\,K^{-1}$, $\kappa_{l,\{1,2\}} = 0.008\ W\,m^{-1}\,K^{-1}$, $\kappa_{r,\{1,2\}} = 0.047\ W\,m^{-1}\,K^{-1}$, $\kappa_{l,\{1,3\}} = 0.020\ W\,m^{-1}\,K^{-1}$, and $\kappa_{r,\{1,3\}} = 0.010\ W\,m^{-1}\,K^{-1}$. For Fig. 3c, $\kappa_{l,\{1,1\}} = 0.039\ W\,m^{-1}\,K^{-1}$, $\kappa_{r,\{1,1\}} = 0.008\ W\,m^{-1}\,K^{-1}$, $\kappa_{l,\{1,2\}} = 0.008\ W\,m^{-1}\,K^{-1}$, $\kappa_{r,\{1,2\}} = 0.040\ W\,m^{-1}\,K^{-1}$, $\kappa_{l,\{1,3\}} = 0.020\ W\,m^{-1}\,K^{-1}$, and $\kappa_{r,\{1,3\}} = 0.010\ W\,m^{-1}\,K^{-1}$. For all cases above, $\kappa_s = 1 \times 10^{-6}\ W\,m^{-1}\,K^{-1}$, $\kappa_{o,\{1,j\}} = \kappa_{pac,\{1,j\}} = 1\ W\,m^{-1}\,K^{-1}$ ($j = 1,2,3$). The second part simulates the multi-temperature maintenance function of AMTC (Fig. 4b, d). To make the case more realistic, we set $\kappa_{o,\{i,j\}} = 0.5\ W\,m^{-1}\,K^{-1}$. The remaining geometry and thermal/structural parameters were the same as those in Supplementary Fig. 5a. The left and right boundary temperatures were set to 1.0 (400 K) and 0.0 (300 K), respectively. We set the convective heat transfer coefficient of the upper and lower boundaries to $10\ W\,m^{-1}\,K^{-1}$. The environmental temperature was −0.0685 (293.15 K). Given the temperature maintenance operations in our daily lives, we should reset the initial temperatures of the system components, as shown in Fig. 4a. Then, we performed transient simulations. We obtained the maximum/average/minimum temperatures of the goods over time (Fig. 4b). Using the "probes" function, we carried out the transient heat fluxes of each temperature control zone (Fig. 4d). For comparison, we replaced the thermal conductivities of the zones marked "a-r" with $0.01\ W\,m^{-1}\,K^{-1}$, $0.005\ W\,m^{-1}\,K^{-1}$, and $0.001\ W\,m^{-1}\,K^{-1}$. Then, the transient characteristics for cases 2–4 were obtained.

Finally, we introduce the transient simulation methods of the multi-temperature maintenance process of the container without AMTC, with AMTC, and with LTCM (Figs. 6b, 7a, b). The structural parameters of the containers with and without AMTC can be found in Supplementary Fig. 21. For the container with AMTC, we set the initial temperatures of each temperature control zone, which includes the package structure (foam cover and glassware) and storage space (goods and the above air), as: $T_① = 325.13\,K$, $T_② = 321.94\,K$, $T_③ = 318.47\,K$, $T_④ = 311.92\,K$, $T_⑤ = 306.97\,K$, $T_⑥ = 300.75\,K$, $T_⑦ = 291.16\,K$, $T_⑧ = 287.52\,K$, $T_⑨ = 284.39\,K$. We set the initial temperatures of the mobile heat and cold sources to 353.15 K and 266.15 K, respectively. We used the "phase-change materials" to simulate the phase transition process in the mobile heat and cold sources. The transition interval was 5 K. The thermal properties of PCMs are shown in Supplementary Table 2. The thermal conductivity, density, specific heat capacity, and specific heat rate of the water were $0.6\ W\,m^{-1}\,K^{-1}$, $998.2\ kg\,m^{-3}$, $4183\ J\,Kg^{-1}\,K^{-1}$, and 1, respectively. The thermal properties of air were extracted from COMSOL Multiphysics. The remaining thermal properties of the system components can be found in Supplementary Table 3. We used the "convection-enhanced thermal conductivity" to simplify the convection heat transfer of the air in the container. We used the "vertical cavity model." The height was 0.152 m, and the width was 0.34 m. We neglected the radiation heat transfer in the container. We set the convective heat transfer coefficient outside the container to $10\ W\,m^{-2}\,K^{-1}$. The environmental temperature was 294.15 K. We set the temperature of the container's face in contact with the ground to

294.15 K. We then performed a transient simulation. The simulation results of the simulacra's temperature in each temperature control zone are shown in the lower subgraph of Fig. 6b, which is consistent with the experimental result. We then showed the temperature distributions and the heat flux analysis in the lower subgraph of Fig. 7a, b, respectively. For the container without AMTC, the initial temperatures of each temperature control zones were set as: $T_① = 325.55$ K, $T_② = 322.11$ K, $T_③ = 318.50$ K, $T_④ = 310.03$ K, $T_⑤ = 305.55$ K, $T_⑥ = 299.61$ K, $T_⑦ = 290.44$ K, $T_⑧ = 287.17$ K, $T_⑨ = 284.42$ K. The initial temperatures of the remaining parts were 294.15 K. The settings for the outer boundary were the same as those of the container with AMTC. For the "convection-enhanced thermal conductivity," we adjusted the Nusselt (Nu) number to make the simulation result of goods' temperature versus time consistent with the experimental result (upper subgraph of Fig. 6b). We found the appropriate Nu number to be around 65. We then obtained the corresponding temperature distributions and heat flux analysis of the container without AMTC, as shown in the upper subgraphs of Fig. 7a, b, respectively. Next, we introduce the simulation methods for the container with LTCM. Its structural parameters were similar to those of the container with AMTC. The height of the low thermal conductivity material used to replace the mobile heat and cold sources was changed to 245 mm. The remaining structural parameters can be found in the container with AMTC. The initial temperature of each temperature control zone was set as: $T_① = 326.27$ K, $T_② = 323.21$ K, $T_③ = 320.12$ K, $T_④ = 311.34$ K, $T_⑤ = 306.11$ K, $T_⑥ = 299.58$ K, $T_⑦ = 291.43$ K, $T_⑧ = 288.96$ K, $T_⑨ = 286.64$ K. The initial temperature of the remaining components was set to 294.15 K. The settings for the outer boundary were the same as those of the container with AMTC. The results were shown in the middle subgraphs of Figs. 6b and 7a, b.

## Fabricate methods of mobile heat and cold sources

The relationship between the mobile heat and cold sources and the 2-D system is depicted in Supplementary Note 11. We selected stearic acid (provided by Sinopharm Chemical Reagent Co., Ltd., China) and distilled water to fabricate the heat and cold sources, respectively. The performance of these heat and cold releases is demonstrated in Supplementary Fig. 15. These results illustrate that the phase transition processes of both substances are stable and favorable for use as mobile heat and cold sources. To control their shapes and prevent leakage of the PCMs in their liquid states, we encased them in high-density polyethylene boxes of suitable dimensions (Supplementary Fig. 16a). For the thermal properties of the two PCMs, one can refer to Supplementary Table 2. Detailed instructions regarding the fabrication, preparation, and installation of mobile heat & cold sources can be found in Supplementary Note 13.2. The consistent temperature gradient established by PCM A and PCM B (Fig. 6d; test points: A1-A3 and B1-B3) provides evidence that this approach is well-suited for a multi-temperature maintenance container.

## Fabricate methods of a multi-temperature maintenance container

We initially set the dimensions of the temperature control zones in line with the sizes of the goods. Select a commercial thermal insulating container of an appropriate size for their storage. Leave sufficient space for the integration of a multi-temperature control system. Thus, the structural parameters of the system ($d_{\epsilon,\{i,j\}}$) can be established. All the thermal parameters ($\kappa_{\epsilon,\{i,j\}}$) can be calculated following the approaches in Supplementary Notes 4 and 5. Then, we chose suitable actual materials (commercial isotropic materials) to fabricate the system (Supplementary Note 11.3). We assessed the multi-temperature control performance of the system using finite element simulations. The minimal discrepancy between simulation results and the required one implies that this method is fit for purpose (Supplementary Fig. 19c). Finally, we positioned the system in specific locations to achieve multi-temperature maintenance for the goods stored within

the temperature control zones (Fig. 5a). Additional details can be found in Supplementary Notes 11 and 13.

## Preparation of the water with different initial temperatures

The fundamental concept behind preparing water with distinct initial temperatures is to blend hot or cold water with water of normal temperature. Supplementary Note 13.1 provides an in-depth method for calculating the proportion between the volumes of hot/cold water and the normal-temperature water for each piece of glassware (Supplementary Fig. 24). The minor deviations between the experimental results and the required values imply that this method is indeed practicable (Supplementary Fig. 25).

## Multi-temperature maintenance test

We utilized T-type thermocouples, a data collector (Keithley 2700, Tektronix Inc., America), and a computer to conduct tests and gather temperature data. The reference positions for the test points of the thermocouples can be found in Supplementary Fig. 21. The comprehensive steps of the experimental procedure are detailed in Supplementary Note 13 (please see Supplementary Fig. 23).

## Characterization of the multi-temperature maintenance performance

We defined the temperature variation rates of the goods stored in the container with AMTC, without AMTC, and with LTCM as follows:

$$\eta_{\gamma,\tau,u,\omega} = \frac{T_{\gamma,\tau,u,\omega} - T_{\gamma,\tau,u,\omega}|_{\tau=0}}{T_{\gamma,\tau,u,\omega}|_{\tau=0}} \times 100\%, u = \{1,2,3\}. \quad (12)$$

In this equation, $\eta_{\gamma,\tau,u,\omega}$ ($T_{\gamma,\tau,u,\omega}$) signifies the temperature variation rates (temperature) of the goods in the temperature control zone $\gamma$ at characteristic time $\tau$ in device $u$ under $\omega^{th}$ test; $\tau = \{0.25$ h, 0.50 h, 0.75 h, 1.00 h, 1.25 h, 1.50 h, 1.75 h, 2.00 h$\}$; $u = 1$, $u = 2$, and $u = 3$ represent the device as the container with AMTC, without AMTC, and with LTCM; $\omega = \{1,2,3,4,5\}$. Based on the five test data, we calculated average temperature variation rates as

$$\left\langle \eta_{\gamma,\tau,u} \right\rangle = \frac{\sum_{\omega=1}^{5} \eta_{\gamma,\tau,u,\omega}}{5}, \quad (13)$$

while the dispersion degree of $\eta_{\gamma,\tau,u,\omega}$ was represented by the experimental standard deviation

$$\sigma_{\gamma,\tau,u} = \sqrt{\frac{\sum_{\omega=1}^{5} \left( \eta_{\gamma,\tau,u,\omega} - \left\langle \eta_{\gamma,\tau,u} \right\rangle \right)^2}{4}}. \quad (14)$$

The lower the $\left\langle \eta_{\gamma,\tau,u} \right\rangle$, the better the multi-temperature maintenance performance.

## Calculation of the average initial temperature of the simulacra stored in each temperature control zone

The average initial temperature of the simulacra stored in temperature control zone $\gamma$ was calculated by

$$\left\langle T_{\gamma,in} \right\rangle = \frac{\sum_{\omega}^{5} T_{\gamma,\tau,u,\omega}|_{\tau=0,u=1} + \sum_{\omega}^{5} T_{\gamma,\tau,u,\omega}|_{\tau=0,u=2} + \sum_{\omega}^{5} T_{\gamma,\tau,u,\omega}|_{\tau=0,u=3}}{15}. \quad (15)$$

## Characterization of the effect of AMTC

We defined the following alterations in the magnitudes of multi-temperature variation as follows:

$$\zeta_{\gamma,\tau,1} = \frac{|\langle \eta_{\gamma,\tau,1} \rangle| - |\langle \eta_{\gamma,\tau,2} \rangle|}{|\langle \eta_{\gamma,\tau,2} \rangle|} \times 100\%$$
$$\zeta_{\gamma,\tau,2} = \frac{|\langle \eta_{\gamma,\tau,1} \rangle| - |\langle \eta_{\gamma,\tau,3} \rangle|}{|\langle \eta_{\gamma,\tau,3} \rangle|} \times 100\%. \tag{16}$$

The higher the magnitude of $\zeta_{\gamma,\tau,1}$ ($\zeta_{\gamma,\tau,2}$), the greater the improvement in multi-temperature maintenance performance when moving from the container without AMTC (the container with LTCM) to the container with AMTC. As this improvement stems from the application of AMTC, the higher the degree of improvement, the more pronounced the enhancement of multi-temperature maintenance performance by AMTC.

## Reporting summary

Further information on research design is available in the Nature Portfolio Reporting Summary linked to this article.

## Data availability

The data generated in this study are provided in the Source Data file. Source data are provided with this paper.

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

## Acknowledgements

We thank Professor Xiaoshi Qian for his constructive suggestions on how to improve the manuscript and thank Professor Chunzhen Fan, Dr. Yong Gao, Mr. Jinrong Liu, and Mr. Fubao Yang for their beneficial discussions and help. J.H. and X.Z. acknowledge financial support from the National Natural Science Foundation of China under Grant No. 12035004 and from the Science and Technology Commission of Shanghai Municipality under Grant No. 20JC1414700. X.X. acknowledges financial support from the National Key R&D Program of China (2022YFC3005800).

## Author contributions

X.Z. and J.H. proposed the project; J.H. supervised the project; X.Z. performed theoretical analysis, finite element simulations, and experiments; X.Z. analyzed the data, performed visualization, and wrote the original manuscript with input from all co-authors. X.Z., X.X., and J.H. reviewed and edited the manuscript.

## Competing interests

The authors declare no competing interests.
