## [Peer Review File · Nature Communications]

Adaptive multi-temperature control for transport and storage containers enabled by phase change materialsREVIEWER COMMENTS

Reviewer #1 (Remarks to the Author):

What are the noteworthy results?

Multi-temperature thermal energy storages for the cold chain are surely worth of investigations. There is the need to develop smart, compact and more efficient solutions. From this point of view, the multi-temperature systems can become an interesting solution.

Will the work be of significance to the field and related fields? How does it compare to the established literature? If the work is not original, please provide relevant references. Does the work support the conclusions and claims, or is additional evidence needed?

This work has a rather low novelty. There are plenty of papers dealing with the multiple PCMs. The idea is not new. The multi-terrace idea is not revolutionary or ground-breaking. Here a list of papers: <https://doi.org/10.1115/HT2022-86174> ; <https://doi.org/10.1016/j.jfoodeng.2020.110286> ; [10.37934/arfmts.78.1.6078](https://doi.org/10.37934/arfmts.78.1.6078) ; <https://doi.org/10.1016/j.enconman.2014.03.047> .

I could have included at least other 10 papers, which have already investigate the topic.

All in all, I cannot recommend the paper for publication in Nature communications because it lacks of the needed inspiring novelty. It is, at least from my perspective, an excellent work for other more applied journals.

Reviewer #2 (Remarks to the Author):

Reliable transportation of life necessities while ensuring high safety and efficiency is a critical social challenge, particularly during epidemics like COVID-19. The key issue in achieving this goal is the heat transfer among life necessities with varying temperature requirements. Therefore, the adoption of adaptive multi-temperature control is essential. In industrial applications like cold chain logistics, liquid-solid phase change materials (PCMs) have greatly benefited transportation. The large latent heat during phase transitions provided by PCMs can prevent the life necessities inside the container from being affected by the outside environment. As a result, this technology has been widely employed. However, due to materials' physicochemical properties, PCMs' fixed phase transition temperatures ask researchers to develop more high-performance composite PCMs that meet various life necessities' temperature requirements, which requires a significant investment. Besides, using PCMs to control various life necessities with varying temperature requirements may decrease space utilization since multiple PCMs need to be prepared and placed in the corresponding areas.

The authors clearly explained the social challenges related to reliable transportation of life necessities and a comprehensive introduction to the advanced multi-temperature control technology and their obstacles. As a response, they proposed an adaptive multi-temperature metaterrace (AMTM) approach that can realize excellent multi-temperature control under complex temperature interactions. To demonstrate the potential of AMTM in practical applications, the authors fabricated a multi-temperature maintenance container based on AMTM and liquid-solid phase transitions of stearic acid and distilled water. It exhibited excellent multi-temperature maintenance performance for the inside simulacra with different initial temperatures. The innovative design in this work can be considered a combination of thermal metamaterials and phase change technology. It implies that the heat and cold sources for AMTM can be further improved by the rapid development of phase change technology, leading to better multi-temperature maintenance performance in the future. The findings and results provide a general framework for designing an efficient multi-temperature control system for objects with different shapes, numbers, phases, thermal properties, etc., which benefits industrial applications. Since the

need to ensure the reliability of objects with varying temperature requirements exists in various fields, such as biological science, energy science, and applied physics, this work is of general interest and suitable for publication in this journal. The following are some suggestions.

(1) In Fig. 1c, the authors presented an exemplary schematic for the phase transition processes of PCMs with heat and cold storage capability. It is known that there might exist supercooling in heat release and cold storage processes. Although this defect can be optimized by various methods in industrial applications, such as adding nucleating agents and applying the "cold finger" method, the authors are suggested to give some explanation in figure captions to make it more detailed.

(2) In "Conclusions," the author wrote "With the fast development of high-performance composite PCMs (phase change materials), the energy densities and the thermal properties of the mobile heat and cold sources can be enhanced, further improving the multi-temperature maintenance performance" and cited two research papers and two review papers related to advanced phase change technologies. It is suggested to add more research related to high-performance composite PCMs, which can act as potential candidates for the mobile heat and cold sources of the multi-temperature maintenance container. It is beneficial for future optimizations and industrial applications.

(3) In the section "Finite element simulations" of "Methods," the author used a generic term in heat transfer, "Nu number." Using the "Nusselt number" when it first appears is suggested.

(4) After reading the Supplementary information carefully, it is found that the size of the glassware is around 80 mm×80 mm×80 mm. This operation is reasonable and sufficient to demonstrate the effect of AMTM on multi-temperature maintenance for practical applications. On the other hand, a larger storage space might be more attractive for utilizers. The authors have given a general theoretical analysis and comprehensive finite element simulations to exhibit the universality of AMTM. A brief discussion of the AMTM with larger storage space in the corresponding Supplementary material might be beneficial.

Reviewer #3 (Remarks to the Author):

Overall very interesting concept. If this is to be realized in real life, the packing of the goods in the container would then need to be prudently performed so as not to break the cascade of the required temperature levels. More detailed comments are shown below.

The use of Covid as motivation is slightly outdated, please generalize this argument.

With the AMTM concept, the cascading of temperature level is very important. This temperature level will depend on the thermal mass, thermal inertia, c_p , initial temperature level of the goods. Please exemplify how this can be optimized for each of the simulacra.

Page 3, the authors mentioned "supercooling and phase separation" as inherent defects, increasing number of research are showing that the supercooling can actually be used for the benefit of certain applications, could the authors develop on this idea for the AMTM?

The reviewer understood that the simulation has been validated against the experimental work, please do provide more information on the experimental rig as well as photographs in the main article.

By increasing the thermal insulation around each of the simulacra, the temperature levels may also be maintained. Please benchmark PCM heat source/sink against pure insulating materials. Are there economic benefit and environmental gain for the same technical outcome?

The arrangement of the goods (simulacra here) needs to be properly placed. Now imagine a wrongly placed simulacrum, what thermal impact would this bring to the system? Please quantify it for a few chosen cases.

Response to the Reviewers

Dear reviewers, thanks for your comments and suggestions. The following are the detailed responses point to point. For clarity, all reference numbers in this reply correspond strictly to the manuscript.

Reviewer #1 (Remarks to the Author):

1. What are the noteworthy results?

Response:

Thank you for your question. This study presents the adaptive multi-temperature metaterrace (AMTM), an approach for efficient multi-temperature control for practical applications. Our work has yielded several noteworthy results:

(1) AMTM enables high-performance multi-temperature control, making it suitable for a wide range of objects with varying shapes, numbers, phases, thermal properties, and sizes. The underlying theoretical model supports unlimited temperature control zones, accommodating any temperature demand between a pair of heat and cold sources.

(2) The key mechanism behind AMTM's multi-temperature control lies in the joint temperature control of multiple zones under a temperature difference provided by a pair of cold and heat sources. Unlike existing technologies that control each zone separately with thermal insulation in between, AMTM exploits heat transfer among objects with different temperature requirements, achieving efficient multi-temperature control.

(3) AMTM exhibits practical value and can be easily implemented. To demonstrate its practical application, we developed a multi-temperature maintenance container for storing diverse life necessities with varying temperature requirements, integrating AMTM with existing phase change technologies. The experimental results proved the

feasibility of AMTM in real-world scenarios.

(4) From an application perspective, AMTM overcomes two significant challenges in advanced multi-temperature control based on phase change technologies: the inherent heat transfer among life necessities with different temperatures and the physicochemical properties of phase change materials (PCMs). It eases the difficulty of implementing efficient multi-temperature control.

2. Multi-temperature thermal energy storages for the cold chain are surely worth of investigations. There is the need to develop smart, compact and more efficient solutions. From this point of view, the multi-temperature systems can become an interesting solution. Will the work be of significance to the field and related fields?

Response:

Multi-temperature control plays a crucial role in various fields, including life necessities transportation¹, cold chain logistics²⁻⁴, building environments⁵, battery thermal management⁶, and thermal energy storage⁷⁻¹¹. Life necessities transportation, in particular, presents unique challenges due to its broader temperature range compared to existing cold chain logistics. However, multi-temperature control has been extensively developed in cold chain logistics, making it a suitable example for our discussion.

In the context of cold chain logistics, advanced technologies for multi-temperature control typically rely on multiple PCMs to regulate different zones, often using one PCM per zone⁴. However, directly using this approach to life necessities transportation faces two obstacles.

Firstly, the properties of the multiple PCMs must be excellent, including suitable phase change temperatures, good thermal properties, and stable thermal, chemical, and cycle stability^{19-21,30,32,33}. Pure PCMs, in their natural state, may have various defects

such as supercooling and phase separation³⁴⁻³⁸. Consequently, a substantial investment is required to develop composite PCMs with suitable phase change temperatures to meet various temperature requirements. This is supported by existing research on the development of high-performance composite PCMs. Take Ref. 35 as an example, the researchers should solve the supercooling problems first and then ensure the remaining performances are stable, which requires plenty of experiments.

Secondly, the current approach controls each temperature control zone using one PCM and utilizes low thermal conductivity materials to minimize heat transfer between zones. However, this method does not eliminate heat transfer between zones, especially when the temperature difference between life necessities is significant. In such cases, additional energy may be required to compensate for the temperature interaction's influence on the multi-temperature control performance. This phenomenon has been observed in related research on multi-temperature insulating box⁴, where a temperature deviation exists in normal temperature storage spaces due to temperature influence from neighboring storage spaces with different temperatures.

To overcome these obstacles, we introduced the concept of AMTM, which has yielded noteworthy results as mentioned earlier. AMTM combines the principles of thermal metamaterials and phase change technologies, allowing for the control of heat flow through elaborative thermal and structural design. Specifically, for the example of reliable life necessities transportation requiring efficient multi-temperature control, two types of PCMs with high and low phase change temperatures can act as mobile heat and cold sources in AMTM, which can be selected from existing research on high-performance PCMs. The remaining components of AMTM can be fabricated using common commercial materials. Consequently, AMTM offers a promising solution for achieving efficient multi-temperature control.

Our study involved a comprehensive exploration of the fundamental theory behind

AMTM, including simulation verification, as well as the development of practical engineering samples that underwent thorough performance evaluations. This framework establishes a foundation for realizing efficient multi-temperature control in practical applications. It is important to note that the demand for efficient multi-temperature control extends beyond life necessities transportation, and the introduction of heat flow regulation through AMTM may have potential benefits in other domains such as cold chain logistics, building environments, and thermal energy storage. By employing the concept of heat flow regulation in AMTM, the requirement for PCMs or other heat and cold sources can be significantly reduced, which is advantageous for practical applications.

This work combined the concept of heat flow regulation in thermal metamaterials with phase change technologies. It addresses critical social challenges related to the reliable transportation of life necessities. In the revised "Discussion," we analyzed the contribution of this work on physical and engineering sciences and its potential value in life science for controlling microbial activity. To the best of our knowledge, the idea of AMTM and the results presented in this study have not been previously reported, underscoring their significant importance for the field and other related areas with multi-temperature requirements.

Please refer to our response to question 3 for more detailed comparisons between our present work and the established literature.

3. How does it compare to the established literature? If the work is not original, please provide relevant references.

Response:

We have conducted a comparison between our present work and the related literature to further highlight the value of our research in the "Discussion" section.

As mentioned above, the application of AMTM in life necessities transportation in this work can be viewed as a combination of thermal metamaterials and phase change technology. We have incorporated relevant established works into our manuscript. Taking into account your suggestions, we compared our present work to related established works as follows.

In the field of thermal metamaterials, Ref. 41 introduced an approach for temperature maintenance through elaborate heat flow regulation. Regardless of the changing temperature difference between the two ends of the system, the temperature of the functional region can always be maintained constant with zero energy input. This intriguing mechanism provides a deep understanding of temperature maintenance by controlling the net heat flow across the temperature control zone, rather than relying on materials with extremely low thermal conductivity coupled with PCMs as in existing technologies. Building on the core idea of heat flow regulation, our present work developed the AMTM approach, which enables efficient multi-temperature control. Compared to Ref. 41, our present work extends the application of temperature maintenance from two-dimensional to three-dimensional systems, from one zone to multiple zones in any number, and from laboratory experimental samples to engineering samples of specific storable objects with movable capacity. This addresses practical societal challenges in life necessities transportation. The comprehensive framework for achieving efficient multi-temperature control, including analytical theory, finite element simulation, and engineering sample design and performance testing, guides the practical application of thermal metamaterials.

In the realm of phase change technology, our work expands its application prospects. As mentioned earlier, the operation of using one PCM per zone faces two obstacles. Our work leverages the temperature control capability during liquid-solid phase transitions to create a constant temperature difference. Subsequently, we

introduce AMTM to create multiple temperature-controllable zones for storing life necessities with varying temperature requirements. This approach reduces the PCM requirements and enhances the efficiency of multi-temperature control, alleviating the challenges associated with implementing efficient multi-temperature technology in the transportation of daily necessities. Importantly, all existing high-performance composite PCMs with phase change temperatures that satisfy the requirements of mobile heat and cold sources can be considered for use in AMTM for life necessities transportation, thereby saving the investment required for PCM research. From the detailed functional point, compared with three temperature-controllable zones realized in the existing advanced technologies in cold chain logistics, nine temperature-controllable zones realized in our present work confirm that the core idea of heat flow regulation in AMTM makes a great contribution to efficient multi-temperature control for practical applications.

In addition, in the "Introduction," we have cited the established literature related to multi-temperature control to expand our background. Compared to these kinds of research, the novelty of our work lies in the concept of AMTM, which only requires a pair of heat and cold sources to control multiple temperature zones, rather than one heat or cold source per zone. To the best of our knowledge, this idea has not been proposed or realized previously.

4. Does the work support the conclusions and claims, or is additional evidence needed?

Response:

The work supports the conclusions and claims. For the main conclusion, we listed the evidence point by point corresponding to the response to question 1.

(1) The analytical theory for multi-temperature control was constructed, as shown in Methods and Supplementary Note 1. The multi-temperature control performance for

storing various life necessities was comprehensively tested, as shown in Figs. 2-4 and Supplementary Figs. 2, 5-12.

(2) Based on the mechanism of AMTM, the high efficiency was revealed, compared with the existing technologies using low thermal conductivity materials and phase change technologies, as shown in Figs. 3, 4 and Supplementary Fig. 14.

(3) Corresponding devices for realizing AMTM for life necessities were fabricated by combining with phase change technologies, as shown in Fig. 3 and Supplementary Fig. 21.

(4) As shown in Fig. 1d, AMTM takes advantage of temperature interactions and has no strict temperature requirements for heat and cold sources. Therefore, it simultaneously overcomes the two overhead obstacles of the existing multi-temperature control technology based on multiple PCMs.

For additional evidence, all the Reviewers gave various beneficial suggestions. We have tried our best to improve the manuscript following these suggestions.

Moreover, we noted that the format of Nature Communications has a “Discussion” rather than “Conclusions.” So, we replaced the original section “Conclusion” with “Discussion.” In this part, we make some extended discussions or claims of the AMTM's further application in other fields. The related references were cited and the corresponding analyses were performed to support our claim.

5. This work has a rather low novelty. There are plenty of papers dealing with the multiple PCMs. The idea is not new. The multi-terrace idea is not revolutionary or ground-breaking. Here a list of papers: <https://doi.org/10.1115/HT2022-86174>; <https://doi.org/10.1016/j.jfoodeng.2020.110286>; 10.37934/arfmts.78.1.6078; <https://doi.org/10.1016/j.enconman.2014.03.047>. I could have included at least other 10 papers, which have already investigate the topic. All in all, I cannot recommend the

paper for publication in Nature communications because it lacks of the needed inspiring novelty. It is, at least from my perspective, an excellent work for other more applied journals.

Response:

Temperature control plays a crucial role in both industrial applications and daily life. With the aim of energy conservation and emission reduction, thermal energy storage and phase change temperature control based on multiple PCMs have been adopted as advanced technologies in various fields. As a result, several studies have focused on using multiple PCMs as temperature control sources for multiple zones.

Existing research in this area typically employs a single PCM to control the temperature of one zone and repeats this process for each additional zone. The four references you mentioned mainly revolve around two key areas: thermal energy storage and cold chain logistics. Link 1 (10.1115/HT2022-86174) explores the enhancement of energy efficiency and operational performance of a multiple PCMs system by incorporating machine learning techniques. Link 2 (10.1016/j.jfoodeng.2020.110286) presents a study on achieving multi-temperature control in three storage spaces of cold chain logistics using two composite PCMs with suitable phase change temperatures. The temperatures of the two storage spaces are directly controlled by corresponding PCMs, while the temperature of the remaining storage spaces is maintained at a normal temperature using insulating materials to mitigate the temperature effects from other storage spaces. Links 3 (10.37934/arfmts.78.1.6078) and 4 (10.1016/j.enconman.2014.03.047) share similar engineering principles to link 1, focusing on cascade thermal storage to enhance system efficiency. Multi-temperature control requirements exist in various fields, such as building environments⁵ and battery thermal management⁶. To enrich the research background, we have added more relevant references to the "Introduction" section (Ref. 1-11).

Among these approaches to multi-temperature control, a common characteristic is the use of one heat or cold source to control the temperature of each zone. For example, in cold chain logistics and thermal energy storage, one PCM is assigned per zone. However, as mentioned earlier, this method faces two main challenges. To address these challenges, we have developed a novel approach called AMTM. AMTM enables multi-temperature control of any number of zones using just a pair of heat and cold sources. In the context of life necessities transportation, we utilize two types of PCMs with high and low phase change temperatures as the heat and cold sources, respectively. **This means that our approach for life necessities transportation only requires two PCMs, as opposed to multiple PCMs (more than two) utilized in existing research, to meet various temperature requirements. In other fields requiring multi-temperature control, PCMs can be replaced by other suitable heat and cold sources, thereby achieving efficient multi-temperature control using similar methods. Consequently, our work can guide the reduction in the number of PCMs (or other suitable heat and cold sources), which is advantageous for practical applications.**

To clarify the discussion above, we illustrate an example using life necessities transportation. Let us assume there are nine life necessities with different temperature requirements that need to be controlled. The following figure presents a clear comparison between AMTM and existing advanced technologies based on multiple PCMs. It is evident that the implementation of AMTM significantly reduces the number of required PCMs. The other component of AMTM can be conveniently fabricated using readily available materials. For more detailed explanations, please refer to Supplementary Note 12.

Our work focuses on addressing a specific social problem in the transportation of life necessities, combining the fundamental advantages of physics with the practical value of engineering science. Building upon previous responses, the novelty of our work can be summarized as follows:

1. We introduce a novel and general approach, termed AMTM, which utilizes analytical theory to achieve efficient multi-temperature joint control for multiple zones using only a pair of heat and cold sources.

2. By integrating AMTM with phase change technologies, we successfully fabricate a multi-temperature maintenance container with nine temperature-controllable storage spaces for the transportation of life necessities. This serves as a practical application example to guide future implementations.

3. In comparison to existing advanced methods, our approach offers advantages

such as easy implementation and a greater number of temperature control zones.

To the best of our knowledge, this efficient and general method covers both fundamental and applied research that has not been reported previously. As mentioned in the "Discussion" section, our work not only combines the fundamental advantages of physics with the practical value of engineering science but also has the potential to contribute to efficient microbial activity control in life sciences. It makes our work attractive to multiple disciplines such as physics, engineering, and life sciences. We believe that the improvements made based on the valuable suggestions from the reviewers will greatly enhance the contribution of our work to Nature Communications.

Reviewer #2 (Remarks to the Author):

Reliable transportation of life necessities while ensuring high safety and efficiency is a critical social challenge, particularly during epidemics like COVID-19. The key issue in achieving this goal is the heat transfer among life necessities with varying temperature requirements. Therefore, the adoption of adaptive multi-temperature control is essential. In industrial applications like cold chain logistics, liquid-solid phase change materials (PCMs) have greatly benefited transportation. The large latent heat during phase transitions provided by PCMs can prevent the life necessities inside the container from being affected by the outside environment. As a result, this technology has been widely employed. However, due to materials' physicochemical properties, PCMs' fixed phase transition temperatures ask researchers to develop more high-performance composite PCMs that meet various life necessities' temperature requirements, which requires a significant investment. Besides, using PCMs to control various life necessities with varying temperature requirements may decrease space utilization since multiple PCMs need to be prepared and placed in the corresponding areas.

The authors clearly explained the social challenges related to reliable transportation of life necessities and a comprehensive introduction to the advanced multi-temperature control technology and their obstacles. As a response, they proposed an adaptive multi-temperature metaterrace (AMTM) approach that can realize excellent multi-temperature control under complex temperature interactions. To demonstrate the potential of AMTM in practical applications, the authors fabricated a multi-temperature maintenance container based on AMTM and liquid-solid phase transitions of stearic acid and distilled water. It exhibited excellent multi-temperature maintenance performance for the inside simulacra with different initial temperatures. The innovative design in this work can be considered a combination of thermal metamaterials and phase change technology. It implies that the heat and cold sources for AMTM can be further improved by the rapid development of phase change technology, leading to better multi-temperature maintenance performance in the future. The findings and results provide a general framework for designing an efficient multi-temperature control system for objects with different shapes, numbers, phases, thermal properties, etc., which benefits industrial applications. Since the need to ensure the reliability of objects with varying temperature requirements exists in various fields, such as biological science, energy science, and applied physics, this work is of general interest and suitable for publication in this journal. The following are some suggestions.

Response:

We are very grateful for your comments and suggestions. All the comments have been responded to point by point as follows.

1. In Fig. 1c, the authors presented an exemplary schematic for the phase transition processes of PCMs with heat and cold storage capability. It is known that there might exist supercooling in heat release and cold storage processes. Although this defect can

be optimized by various methods in industrial applications, such as adding nucleating agents and applying the "cold finger" method, the authors are suggested to give some explanation in figure captions to make it more detailed.

Response:

We have added some explanations in the figure captions for Fig. 1c. We have explained that the heat release process is ideal and neglects to supercool. The detailed utilization of supercooling has been added in the "Discussion."

2. In "Conclusions," the author wrote "With the fast development of high-performance composite PCMs (phase change materials), the energy densities and the thermal properties of the mobile heat and cold sources can be enhanced, further improving the multi-temperature maintenance performance" and cited two research papers and two review papers related to advanced phase change technologies. It is suggested to add more research related to high-performance composite PCMs, which can act as potential candidates for the mobile heat and cold sources of the multi-temperature maintenance container. It is beneficial for future optimizations and industrial applications.

Response:

Since various PCMs for cold chain logistics have been summarized in review works, high-performance composite PCMs with low phase change temperatures for mobile cold sources can be found in this kind of research. We added some related high-temperature composite PCMs as a candidate for mobile heat sources. Combined with the suggestions of Reviewer 3, the PCMs with stable supercooling and phase change hysteresis has been considered^{35,60-64}. For instance, some alum-based composite PCMs and eutectic composite PCMs with stable supercooling properties and can provide stable heat for the hot end of AMTM were cited^{35,61}. Their safe and non-toxic properties make them candidates for practical use.

3. In the section "Finite element simulations" of "Methods," the author used a generic term in heat transfer, "Nu number." Using the "Nusselt number" when it first appears is suggested.

Response:

We have revised this kind of expression and carefully checked the whole manuscript and Supplementary Notes.

4. After reading the Supplementary information carefully, it is found that the size of the glassware is around 80mm×80mm×80mm. This operation is reasonable and sufficient to demonstrate the effect of AMTM on multi-temperature maintenance for practical applications. On the other hand, a larger storage space might be more attractive for utilizers. The authors have given a general theoretical analysis and comprehensive finite element simulations to exhibit the universality of AMTM. A brief discussion of the AMTM with larger storage space in the corresponding Supplementary material might be beneficial.

Response:

Assuming there exists a requirement to make the multi-temperature control system larger for storing more life necessities. The core problem the users might concern about is when the size is larger, the temperature derivation in the temperature control zone might be increased. We have added a brief discussion in Supplementary Note 14.1. The discussion can be divided into two aspects. The first is the multi-temperature control mechanism analysis of AMTM, while the second is the validations of finite element simulations.

For the first aspect, we perform a brief discussion based on theoretical analysis. Following the description of the conduction heat transfer system in Fig. 1f, the

temperature difference in each temperature control zone can be expressed as

$$\Delta T_{c,\{m,j\}} = \frac{(T_A - T_B)d_{c,\{m,j\}}/\kappa_{c,\{m,j\}}}{\sum_{j^\#=1}^n \sum_{\epsilon=l,c,r} d_{\epsilon,\{m,j^\#\}}/\kappa_{\epsilon,\{m,j^\#\}}}. \quad (\text{R1})$$

From the perspective of dimension, $\Delta T_{c,\{m,j\}}$ is independent of the size of the multi-temperature control system. In the steady state, keeping the ratio of all structural parameters long X -axis unchanged can maintain the original multi-temperature control performance.

For ease of understanding, we set two groups of structural parameters. One group for small size, which is expressed by $d_{\epsilon,\{m,j\}}^A$. Another group is for large size, which is $d_{\epsilon,\{m,j\}}^B$, where $d_{\epsilon,\{m,j\}}^B = 1000d_{\epsilon,\{m,j\}}^A$. Now, let us consider a specific situation. If our existing operation is designed for a small size case when the structural parameters $d_{\epsilon,\{m,j\}}^A$ were determined according to the requirements, following the description in Supplementary Note 5, we can solve all the thermal parameters $\kappa_{\epsilon,\{m,j\}}$. Therefore, the multi-temperature control system for the small-size case is successfully designed. Then, let us consider the large-size case. A simple method is to replace $d_{\epsilon,\{m,j\}}^A$ with $d_{\epsilon,\{m,j\}}^B$. Rewriting Eq. (R1), we have

$$\Delta T_{c,\{m,j\}} = \frac{(T_A - T_B)d_{c,\{m,j\}}^B/\kappa_{c,\{m,j\}}}{\sum_{j^\#=1}^n \sum_{\epsilon=l,c,r} d_{\epsilon,\{m,j^\#\}}^B/\kappa_{\epsilon,\{m,j^\#\}}}. \quad (\text{R2})$$

Considering the relationship between $d_{\epsilon,\{m,j\}}^A$ and $d_{\epsilon,\{m,j\}}^B$, Eq. (R2) can be rewritten to

$$\begin{aligned} \Delta T_{c,\{m,j\}} &= \frac{(T_A - T_B)1000d_{c,\{m,j\}}^A/\kappa_{c,\{m,j\}}}{\sum_{j^\#=1}^n \sum_{\epsilon=l,c,r} 1000d_{\epsilon,\{m,j^\#\}}^A/\kappa_{\epsilon,\{m,j^\#\}}} \\ &= \frac{(T_A - T_B)d_{c,\{m,j\}}^A/\kappa_{c,\{m,j\}}}{\sum_{j^\#=1}^n \sum_{\epsilon=l,c,r} d_{\epsilon,\{m,j^\#\}}^A/\kappa_{\epsilon,\{m,j^\#\}}}. \end{aligned} \quad (\text{R3})$$

The consistency of Eq. (R2) and Eq. (R3) ensures that when the device is scaled up, the effect of multi-temperature control can be maintained in a steady state. This result can be examined by simple two-dimensional finite element simulations. We take one case

in Fig. 2a as an example and enlarge the system size from “mm” to “m”, and performed simulations. The comparison between these two sizes is shown in the following table.

To make it clear, we sort the maximum/average/minimum temperature of each temperature control zone in the following table. It can be seen that the temperature distribution of these two cases is the same, which is of great consistency with the above results.

Case	Temperature (K)	Zone		
		1	2	3
1	Maximum	303.22	285.72	280.22
	Average	303.15	285.65	280.15
	Minimum	303.08	285.58	280.08
2	Maximum	303.22	285.72	280.22
	Average	303.15	285.65	280.15
	Minimum	303.08	285.58	280.08

Moreover, if the users want to further increase the storage space in the above

case of large size under the condition of the constant size of the whole system, they might consider reducing the size of the red and blue zones ($d_{l,\{m,j\}}$ & $d_{r,\{m,j\}}$) in the multi-temperature control system, as shown in Fig. 1f. Although this operation will destroy the structural parameters ratio mentioned above, our theory can solve this problem by redesigning the thermal parameters $\kappa_{\epsilon,\{m,j\}}$, as shown in Supplementary Note 5.

From the application view of life necessities transportation, the transient state of the multi-temperature maintenance performance of the container with AMTM should be further tested. Following the simulation method of Fig. 3e, we enlarged the size of the container from “mm” to “m” and then performed the simulations. The simulation results indicate that within two hours, the temperature of objects in each zone can be well maintained at a constant temperature, as shown in the following figure a.

We further conducted a special case study. The objects in the high-temperature

region were removed and replaced with normal-temperature air, it has been observed that the temperature of the objects in the remaining areas can still be effectively maintained, as shown in figure b above. The reason is that while increasing the size of the container extends the time it takes for the heat source and cold source to regulate the temperature in each zone, it also increases the storage volume for objects in each zone. Consequently, the sensible heat storage capacity is further enhanced, allowing the container to remain unaffected by the external environment over a short period. This demonstrates an exceptional ability to retain multiple temperatures simultaneously. These findings indicate that enlarging the size of a multi-temperature maintenance container does not compromise its performance, making it suitable for various applications.

Reviewer #3 (Remarks to the Author):

Overall very interesting concept. If this is to be realized in real life, the packing of the goods in the container would then need to be prudently performed so as not to break the cascade of the required temperature levels. More detailed comments are shown below.

Response:

We are very grateful for your comments and suggestions. We respond to each suggestion or question point by point as follows.

1. The use of Covid as motivation as slightly outdated, please generalize this argument.

Response:

We have removed the specific mention of "COVID-2019." Following the suggestion of reviewer 1, we have provided a general background in the first paragraph of the "Introduction." The main idea conveyed is that there is a demand for multi-temperature control in various fields. Achieving high efficiency in multi-temperature

control is crucial for the advancement of society and the well-being of humanity. Subsequently, we have focused on the specific problem of multi-temperature control in the transportation of essential goods, using it as an example to introduce our motivation for this work. We believe that the argument has now been broadened and made more general.

2. With the AMTM concept, the cascading of temperature level is very important. This temperature level will depend on the thermal mass, thermal inertia, c_p , initial temperature level of the goods. Please exemplify how this can be optimized for each of the simulacra.

Response:

We refer to the factors mentioned as the sensible heat storage capacity Q_{HS} and cold storage capacity Q_{CS} . Increasing the values of Q_{HS} and Q_{CS} results in larger thermal inertia. To optimize each simulacrum, we propose two strategies:

(1) Increase the specific heat capacity c_p and density ρ of each simulacrum.

(2) Increase the initial temperatures $T_{\gamma,in}$ of the simulacrum whose temperature is higher than the environmental temperature T_E , while decrease $T_{\gamma,in}$ of those lower than T_E , in the high- and low-temperature regions.

For case 1, finite element simulations were performed based on the simulation for the multi-temperature maintenance container. Setting c_p to $1000 \text{ J kg}^{-1} \text{ K}^{-1}$, $2000 \text{ J kg}^{-1} \text{ K}^{-1}$, and $3000 \text{ J kg}^{-1} \text{ K}^{-1}$, the effect of specific heat capacity on the multi-temperature maintenance can be revealed by observing the temperature-time curves of each simulacrum in the container. Similarly, setting ρ to 2000 kg m^{-3} , 3000 kg m^{-3} , and 4000 kg m^{-3} , the effect of density on the multi-temperature maintenance performance could be revealed.

For case 2, we increased the temperatures of simulacra in the high-temperature

region by 1 K, 2K, and 3K, while reducing those in the low-temperature region by 1 K, 2K, and 3K.

We present the simulation results in the figure below. The results demonstrate that increasing the sensible heat or cold storage capacity by enhancing the specific heat capacity and density of each simulacrum has a positive impact on reducing temperature variations. Additionally, adjusting the initial temperature of the simulacrum towards a higher or lower value, respectively, can further extend the effective duration of maintaining multiple temperatures. This approach can be considered as pre-cooling or pre-heating and is utilized to compensate for heat or cooling losses resulting from the initial heat transfer between the simulacrum and the system.

3. Page 3, the authors mentioned “supercooling and phase separation” as inherent defects, increasing number of research are showing that the supercooling can actually be used for the benefit of certain applications, could the authors develop on this idea for the AMTM?

Response:

Based on the points you mentioned, it is indeed recognized in research that supercooling can have beneficial effects in certain applications, such as seasonal thermal energy storage and everyday life utilization^{35,60,61}. When considering the implementation of AMTM in the transportation of essential life support materials, supercooling and phase change hysteresis can impact the solidification of PCM A for heat release and the melting of PCM B for cold storage. For PCM B, if the user does not care about the efficiency of cold storage, then supercooling does not affect the low-temperature boundary T_B of AMTM. The following discussion focuses on PCM A. We can develop two ideas from the perspectives of thermal energy release and storage.

(1) Heat release: Following the AMTM mechanism depicted in Fig. 1f, the solidification process of PCM A is utilized to provide a constant temperature boundary, T_A . Therefore, the requirement for PCM A is not that it should completely eliminate supercooling or phase change hysteresis. Instead, PCM A should be capable of supplying a stable and suitable temperature during the solidification stage. Existing technologies, such as incorporating nucleating agents to create composite materials with stable supercooling, can expand the selection of potential PCM A candidates^{35-38,58}.

(2) Heat storage: Existing technologies that enable stable supercooling, known as seasonal thermal energy storage^{60,61}, can keep PCM A in a stable liquid state for an extended period. By triggering crystallization, it can release heat when required. If this concept is applied in AMTM, it can facilitate the heat storage process of PCM A.

Thanks to various research efforts focused on addressing these challenges, users can now choose suitable PCMs with the appropriate solidification temperature. We have included relevant references and ideas in the "Discussion" section.

4. The reviewer understood that the simulation has been validated against the experimental work, please do provide more information on the experimental rig as well as photographs in the main article.

Response:

In addition to the aforementioned suggestions, we have further enhanced the experimental setup by including detailed photographs of the container with AMTM, the container without AMTM, and the container with low thermal conductivity materials (LTCM). These additions are depicted in Fig. 3b as the experimental rig. Furthermore, to provide a more comprehensive understanding of the experimental phenomena, we have included an infrared image of the container with AMTM, as shown in Fig. 3f.

5. By increasing the thermal insulation around each of the simulacra, the temperature levels may also be maintained. Please benchmark PCM heat source/sink against pure insulating materials.

Response:

We have included the case involving pure insulating materials as you mentioned. Additionally, all experiments were repeated independently five times to ensure reliability. We designated the case with low thermal conductivity material (LTCM) as control group 2 for comparison. See a detailed explanation of the components in the following figure.

The results of the comparison among the case without AMTM, the case with LTCM, and the case with AMTM are depicted in Fig. 3d, e and Fig. 4. See the subfigures marked in red squares in the following figures. From the figures, it is evident that the inclusion of pure insulating materials effectively enhances the performance of multi-temperature maintenance. However, the performance can be further improved by incorporating AMTM.

Editorial Note: Elements of panel (a) below have been redacted as indicated to remove third-party material where no permission to publish could be obtained.

Fig. 3 in the manuscript.

Fig. 4 in the manuscript.

6. Are there economic benefit and environmental gain for the same technical outcome?

Response:

The application of AMTM offers both economic benefits and environmental gains while achieving the desired technical outcome. Let us consider the example of life necessities transportation.

The implementation of AMTM involves two key factors. Firstly, it requires two types of high-performance PCMs. Secondly, common commercial insulating materials are utilized. As these components are selectable, we have the flexibility to choose cost-effective materials based on existing research. This allows us to fabricate the device in a manner that ensures comparable economic benefits to existing advanced technologies.

Moreover, all the components used in AMTM are chosen for their good cycle stability and non-toxic properties, enabling their recycling. This environmentally friendly approach is exemplified in the multi-temperature maintenance container we have fabricated, as depicted in Supplementary Fig. 18.

7. The arrangement of the goods (simulacra here) needs to be property placed. Now imagine a wrongly placed simulacrum, what thermal impact would this bring to the system? Please quantify it for a few chosen cases.

Response:

To investigate this scenario, we conducted corresponding experiments, as shown in the following figure.

In Fig. a, two cases of misplacement are presented. In the first case, the simulacra were misplaced in the high-, middle, and low-temperature regions, respectively. In the

second case, which is more serious, the simulacra were misplaced in the low-temperature region, when they should be in the high-temperature region. Conversely, the simulacra intended for the low-temperature region were mistakenly placed in the high-temperature region.

The experimental results are shown in Fig. b. It can be observed that the effect does not appear to be significant for the first case. However, in the second case, the temperature of the simulacra at the misplaced positions has changed considerably. To provide a quantitative description, we analyzed the temperature variation magnitude of each simulacrum after two hours, and the results are presented in Fig. c. The correct replacement results are indicated by the circles in the figure for reference.

From the analysis, we can conclude that if an object is misplaced within the same temperature region (high temperature/middle temperature/low temperature), it will not have a substantial impact on the temperature maintenance effect. However, if an object is misplaced across different temperature zones, such as when the highest-temperature object and the lowest-temperature object are swapped, it will only affect the temperature maintenance effect of these two misplaced objects and will not impact those of other objects. The extent of this effect primarily depends on the deviation between the preset temperature of the temperature control zone and the temperature of the misplaced object. The deviation is smaller within the same temperature region but larger across different temperature regions.

In practical applications, due to the noticeable temperature difference between the mobile heat and sources, users can easily discern the location of the high-temperature region and the low-temperature region. Therefore, under normal circumstances, misplacements are more likely to occur within the same temperature region. As a result, misplacement does not significantly impact the multi-temperature maintenance effect. The above discussion was added in Supplementary Note 14.3.

Other revisions explanation

In addition to the above modifications, we made some other revisions. The main parts are listed as follows.

- Various elements in Fig. 1 and Fig. 2, such as fish, pizza, etc., were improved to enhance the figure quality.
- Fig. 4c was improved following the “editorial requests.” All the data underlying were shown as the square, circle, and triangle points. In addition, we tried to avoid the simultaneous appearance of red and green as much as possible and changed the original green to yellow or orange.
- The expressions in the manuscript and Supplementary Notes were improved.

Yours sincerely,

Prof. Jiping Huang, Fudan University

REVIEWERS' COMMENTS

Reviewer #2 (Remarks to the Author):

The authors have revised and responded to the previous suggestions and questions. The reviewer has no problem with these contents. In the revised manuscript, the authors have made improvements in the background, experiments, and discussion sections. The addition of an extra control group for experimental testing of multi-temperature control performance is beneficial for comparison. Furthermore, the introduction now provides a broader background by discussing the requirements of multi-temperature systems in various fields.

In the discussion section, the authors have included practical situations such as size enlargement, thermal inertia, and wrong placement, which are useful for real-world applications. Since this work can be considered a combination of thermal metamaterials and phase change technology, the authors have compared and discussed the differences between this work and advanced temperature control methods in these two existing technologies. They have also highlighted the advantages of their approach and presented actual device prototypes along with comprehensive performance tests. Moreover, a further discussion on how to improve the performance of the multi-temperature control in AMTM was included for future consideration. This discussion is beneficial for practical applications. Considering the wide-ranging impacts of multi-temperature control requirements in diverse disciplines like applied physics, engineering science, and life science, this work is suitable for publication in this journal.

Some detailed suggestions are listed below.

1. The reviewer understands that the initial sentence in the abstract has been enhanced to provide a broader context for multi-temperature control. To demonstrate the effectiveness of multi-temperature control, an example highlighting the transportation of essential life necessities has been incorporated. Considering the significance of safety and efficiency emphasized in the original manuscript, it is suggested to incorporate this aspect in the first sentence to underscore the pressing and indispensable nature of multi-temperature requirements in the revised manuscript.
2. In the discussion section, the author states, "Our method relies on careful consideration of the system's thermal and structural parameters rather than conventional thermal insulation methods." The intention behind this statement is likely to highlight the advantage of finely tuned heat flow. To ensure clarity and completeness, it could be beneficial to include the condition of "under a pair of heat and cold sources" and compare it to existing technologies that "use one heat or cold source to control one zone" in this sentence. This addition might provide a comprehensive understanding of the strengths inherent in this work.
3. In the context of future development, the utilization of three pairs of heat and cold sources to mitigate external environmental interference aligns with the established principles of phase change technology in cold chain logistics. It might be beneficial to cite some related references to illustrate the feasibility of the scheme.
4. In the methods section, on page 20, the authors introduced a definition of temperature variation magnitude change $\zeta_{(\gamma, \tau, 2)}$ due to the inclusion of a control group called the container with LTCM. Considering that both $\zeta_{(\gamma, \tau, 1)}$ and $\zeta_{(\gamma, \tau, 2)}$ were utilized to analyze the impact of AMTM in the results section, it may be more appropriate to use plural forms instead of singular forms.

Reviewer #3 (Remarks to the Author):

The work is of high relevance to the society for thermal management during transportation of goods. Most of the comments have been addressed and thorough rebuttals have been provided. The paper is of publication quality if the authors can further include the following two points

1. more advanced sensitivity analysis on different parameters: e.g. cp of the transported goods, time, thermal conductivity of the goods, PCMs
2. generalization of the results with non-dimensional analysis based on thermo-physical properties of the requested transportation goods, transportation time, outdoor ambient temperature, among others so as to decide on the optimal PCM peak melting/freezing temperatures and dimension of each of the compartment.

Response to the Reviewers R2

Dear reviewers, thanks for your comments and suggestions. The following are the detailed responses point to point. For clarity, all reference numbers in this reply correspond strictly to the manuscript.

Reviewer #2 (Remarks to the Author):

The authors have revised and responded to the previous suggestions and questions. The reviewer has no problem with these contents. In the revised manuscript, the authors have made improvements in the background, experiments, and discussion sections. The addition of an extra control group for experimental testing of multi-temperature control performance is beneficial for comparison. Furthermore, the introduction now provides a broader background by discussing the requirements of multi-temperature systems in various fields.

In the discussion section, the authors have included practical situations such as size enlargement, thermal inertia, and wrong placement, which are useful for real-world applications. Since this work can be considered a combination of thermal metamaterials and phase change technology, the authors have compared and discussed the differences between this work and advanced temperature control methods in these two existing technologies. They have also highlighted the advantages of their approach and presented actual device prototypes along with comprehensive performance tests. Moreover, a further discussion on how to improve the performance of the multi-temperature control in AMTM was included for future consideration. This discussion is beneficial for practical applications. Considering the wide-ranging impacts of multi-temperature control requirements in diverse disciplines like applied physics,

engineering science, and life science, this work is suitable for publication in this journal.

Some detailed suggestions are listed below.

Thank you for your suggestions. We address each point sequentially as follows.

1. The reviewer understands that the initial sentence in the abstract has been enhanced to provide a broader context for multi-temperature control. To demonstrate the effectiveness of multi-temperature control, an example highlighting the transportation of essential life necessities has been incorporated. Considering the significance of safety and efficiency emphasized in the original manuscript, it is suggested to incorporate this aspect in the first sentence to underscore the pressing and indispensable nature of multi-temperature requirements in the revised manuscript.

Response:

We have revised the opening sentence in the abstract. In conjunction with the editor's suggestion, this sentence has been amended to: "The transportation of essential items, such as food and vaccines, often requires adaptive multi-temperature control to maintain high safety and efficiency, especially during epidemics or other disasters."

2. In the discussion section, the author states, "Our method relies on careful consideration of the system's thermal and structural parameters rather than conventional thermal insulation methods." The intention behind this statement is likely to highlight the advantage of finely tuned heat flow. To ensure clarity and completeness, it could be beneficial to include the condition of "under a pair of heat and cold sources" and compare it to existing technologies that "use one heat or cold source to control one zone" in this sentence. This addition might provide a comprehensive understanding of the strengths inherent in this work.

Response:

We have revised the corresponding sentence to “Our methodology focuses on an elaborate design of the system's thermal and structural parameters, utilizing a pair of high- and low-temperature PCMs, instead of conventional strategies with one PCM per zone and thermal insulation.”

3. In the context of future development, the utilization of three pairs of heat and cold sources to mitigate external environmental interference aligns with the established principles of phase change technology in cold chain logistics. It might be beneficial to cite some related references to illustrate the feasibility of the scheme.

Response:

We have referred to articles 19, 21, and 70, which delve into the use of PCMs in cold chain logistics. Drawing from these studies, three pairs of heat and cold sources can be set up for the AMTC approach.

4. In the methods section, on page 20, the authors introduced a definition of temperature variation magnitude change $\zeta_{(\gamma,\tau,2)}$ due to the inclusion of a control group called the container with LTCM. Considering that both $\zeta_{(\gamma,\tau,1)}$ and $\zeta_{(\gamma,\tau,2)}$ were utilized to analyze the impact of AMTM in the results section, it may be more appropriate to use plural forms instead of singular forms.

Response:

We have revised the corresponding sentence to plural forms.

Reviewer #3 (Remarks to the Author):

The work is of high relevance to the society for thermal management during transportation of goods. Most of the comments have been addressed and thorough

rebuttals have been provided. The paper is of publication quality if the authors can further include the following two points.

Thank you for your suggestion. We address the two points as follows.

1. More advanced sensitivity analysis on different parameters: e.g. c_p of the transported goods, time, thermal conductivity of the goods, PCMs.

Response:

We have explored the impact of c_p on efficient transportation time in our thermal inertia research, as detailed in Supplementary Note 14.2. The findings indicate that an elevated c_p extends the transportation time, attributed to the augmented sensible heat storage of the goods. The effect of the thermal conductivity of the goods on multi-temperature control is presented in Supplementary Note 9.1. Results underscore that if the effective thermal conductivity of the temperature control zones notably surpasses those of other zones, the thermal conductivity of the goods does not affect the temperature control performance. Furthermore, in Supplementary Note 11.4, we delved into the role of PCMs' temperature distribution in multi-temperature maintenance performance. A key takeaway is the indispensability of temperature uniformity for establishing the high- and low-temperature boundaries for AMTC. Enhancing thermal conductivity, therefore, is favorable for optimized multi-temperature maintenance. A comprehensive sensitivity analysis, weaving in the factors mentioned, holds promise for future innovations. We incorporated a segment in the “Discussion” section to delineate the limitation of our present research and the perspective for further development.

2. Generalization of the results with non-dimensional analysis based on thermo-physical properties of the requested transportation goods, transportation time, outdoor

ambient temperature, among others so as to decide on the optimal PCM peak melting/freezing temperatures and dimension of each of the compartment.

Response:

We have introduced a succinct discussion to determine the melting/freezing temperatures and dimensions of each compartment in Supplementary Note 15.

Supplementary Note 11.4 illustrates that the temperature of the PCMs is crucial for ensuring multi-temperature maintenance performance. Based on Fig. 5c, we offer a foundational analysis to investigate the interrelation among PCM A, PCM B, and the environment from a heat transfer view.

For analytical purposes, we assume the temperature distribution of the PCMs remains uniform during the phase transition process. Consequently, the heat flow from PCM A to PCM B can be articulated as

$$Q_{A \rightarrow B} = \sum_{i=1}^m A_{A \rightarrow B, i} q_i = \sum_{i=1}^m A_{A \rightarrow B, i} \frac{T_{p,A} - T_{p,B}}{\sum_{j=1}^n \sum_{\epsilon=l,c,r} d_{\epsilon, \{i,j\}} / \kappa_{\epsilon, \{i,j\}}} \quad (1)$$

In this equation, $A_{A \rightarrow B, i}$ and q_i represent the heat transfer area and heat flux from PCM A to PCM B on row i , respectively.

Further, the heat flow from PCM A to the environment and that from the environment to PCM B are expressed as

$$\begin{cases} Q_{A \rightarrow E} = A_{A \rightarrow E} \frac{T_{p,A} - T_E}{R_{A \rightarrow E}} \\ Q_{E \rightarrow B} = A_{E \rightarrow B} \frac{T_E - T_{p,B}}{R_{E \rightarrow B}} \end{cases} \quad (2)$$

where $A_{A \rightarrow E}$ and $A_{E \rightarrow B}$ represent the heat transfer areas between PCM A and the environment, and between the environment and PCM B, respectively. For brevity, we characterize the thermal resistance between PCM A (PCM B) and environment as $R_{A \rightarrow E}$ ($R_{E \rightarrow B}$). Subsequently, the effective temperature control times of PCM A and PCM B are

$$\begin{cases} t_A = \frac{L_A}{Q_{A \rightarrow B} + Q_{A \rightarrow E}} \\ t_B = \frac{L_B}{Q_{A \rightarrow B} + Q_{E \rightarrow B}} \end{cases}, \quad (3)$$

where L_A and L_B denote the latent heats of PCM A and PCM B, respectively. In accordance with the AMTC principle, PCM A and PCM B should concurrently offer constant temperature boundaries on the left and right. Thus, the effective temperature control time adopts the minimum value between t_A and t_B

$$t_{\text{eff}} = \text{Min}(t_A, t_B). \quad (4)$$

From Eqs. (1)-(4), one can deduce that diminishing the temperature variance between the phase-change temperatures of PCM A (PCM B) and the environmental temperature, enhancing the thermal resistance from PCM A to PCM B, and magnifying the latent heat can elevate t_{eff} . In synergy with the AMTC principle, selecting PCM A (PCM B) with a phase-change temperature marginally above (below) the utmost (lowest) temperature necessity of the goods, paired with high latent heat, appears beneficial. Given the demand for temperature uniformity, curbing the thermal conductivity in zones $L_{\{i,j\}}$ and $R_{\{i,j\}}$ aids in amplifying t_{eff} . Moreover, increasing the size can augment $d_{\epsilon,\{i,j\}}$, favoring the elevation of thermal resistance between PCM A and PCM B. This also aligns with the anticipated growth in storage space dimensions in real-world applications.

Considering the aforementioned relationships between effective temperature control duration, phase-change temperature, and the system's thermal/structural parameters, future developments might center on opting for high-performance composite PCMs, as endorsed by relevant references, and understanding the impact of thermal/structural parameters of the multi-temperature control system.

Other revisions explanation

In addition to the modifications mentioned above, we have revised the color scheme of both the main text and supplementary materials based on the editor's suggestions. We have also removed or simplified certain image elements that could potentially involve third-party copyright issues while ensuring that this does not impact the presentation of the paper. Furthermore, we have made every effort to enhance the language. Additionally, we have included additional references (42-45) to provide a more comprehensive introduction to the topic of thermal metamaterials.

Yours sincerely,

Prof. Jiping Huang, Fudan University
